# Cohesin-mediated loop anchors confine the locations of human replication origins

Daniel J. Emerson[1,2,3,14], Peiyao A. Zhao[4,14], Ashley L. Cook[1,2,3,14], R. Jordan Barnett[1,2,3], Kyle N. Klein[4], Dalila Saulebekova[5], Chunmin Ge[1,2,3], Linda Zhou[1,2,3], Zoltan Simandi[1,2,3], Miriam K. Minsk[1,2,3], Katelyn R. Titus[1,2,3], Weitao Wang[5], Wanfeng Gong[1,2,3], Di Zhang[6], Liyan Yang[7], Sergey V. Venev[7], Johan H. Gibcus[7], Hongbo Yang[8], Takayo Sasaki[9], Masato T. Kanemaki[10,11], Feng Yue[8], Job Dekker[7,12], Chun-Long Chen[5], David M. Gilbert[4,9] & Jennifer E. Phillips-Cremins[1,2,3,13 ✉]

DNA replication occurs through an intricately regulated series of molecular events and is fundamental for genome stability[1,2]. At present, it is unknown how the locations of replication origins are determined in the human genome. Here we dissect the role of topologically associating domains (TADs)[3–6], subTADs[7] and loops[8] in the positioning of replication initiation zones (IZs). We stratify TADs and subTADs by the presence of corner-dots indicative of loops and the orientation of CTCF motifs. We find that high-efficiency, early replicating IZs localize to boundaries between adjacent corner-dot TADs anchored by high-density arrays of divergently and convergently oriented CTCF motifs. By contrast, low-efficiency IZs localize to weaker dotless boundaries. Following ablation of cohesin-mediated loop extrusion during G1, high-efficiency IZs become diffuse and delocalized at boundaries with complex CTCF motif orientations. Moreover, G1 knockdown of the cohesin unloading factor WAPL results in gained long-range loops and narrowed localization of IZs at the same boundaries. Finally, targeted deletion or insertion of specific boundaries causes local replication timing shifts consistent with IZ loss or gain, respectively. Our data support a model in which cohesin-mediated loop extrusion and stalling at a subset of genetically encoded TAD and subTAD boundaries is an essential determinant of the locations of replication origins in human S phase.

The interphase human genome folds into TADs and nested sub-TADs. TADs were originally defined in first-generation Hi-C and 5C data as megabase (Mb)-scale, self-interacting chromatin segments in which DNA sequences exhibit substantially higher contact frequency within—compared to between—domains[3–6]. Molecular and computational advances over the past decade have resulted in ultrahigh-resolution genome folding maps with substantially improved signal-to-noise ratios[8–11]. Such technical advances have enabled the discovery of fine-grained A/B compartments[8], nested sub-TADs within TADs[7], punctate dot structures indicative of long-range looping interactions[8], and stripes indicative of loop extrusion[12–14]. In light of the critical importance of dissecting the link between specific higher-order chromatin architectural features and genome function, a leading challenge is to classify subtypes of TADs/subTADs in Hi-C maps by their fine-grained structural features. Clearly defining structural classes of TADs/subTADs can in turn facilitate the careful

dissection of each boundary's molecular composition, organizing principles and unique cause-and-effect relationship across a range of genome functions.

Here we ascertain the functional link between distinct structural classes of TADs/subTADs and DNA replication. Replication initiates from tens of thousands of origins licensed in excess across the human genome in telophase and throughout G1 (refs.[1,2]). A small proportion of licensed origins subsequently fire in orchestrated temporal waves during S phase[2]. It is established that origins fire at one or more sites chosen stochastically within ≈40 kb regions (IZs)[15–17]. Nevertheless, a consensus sequence encoding origin or IZ placement has not been definitively identified in humans. Waves of early and late replication correlate with A and B compartments, respectively, and the temporal transitions from early to late replication can in some cases align with TAD boundaries[3,18,19]. However, the role of fine-scale genome folding patterns during interphase (such as loops, subTADs and TADs

[1]Department of Bioengineering, University of Pennsylvania, Philadelphia, PA, USA. [2]Epigenetics Institute, Perelman School of Medicine, University of Pennsylvania, Philadelphia, PA, USA. [3]Department of Genetics, Perelman School of Medicine, University of Pennsylvania, Philadelphia, PA, USA. [4]Department of Biological Science, Florida State University, Tallahassee, FL, USA. [5]Institut Curie, PSL Research University, CNRS UMR3244, Dynamics of Genetic Information, Sorbonne Université, Paris, France. [6]Children's Hospital of Pennsylvania, Philadelphia, PA, USA. [7]University of Massachusetts Chan Medical School, Worcester, MA, USA. [8]Department of Biochemistry and Molecular Genetics, Feinberg School of Medicine, Northwestern University, Chicago, Illinois, USA. [9]San Diego Biomedical Research Institute, San Diego, CA, USA. [10]Department of Chromosome Science, National Institute of Genetics, Research Organization of Information and Systems (ROIS), Mishima, Japan. [11]Department of Genetics, The Graduate University for Advanced Studies (Sokendai), Mishima, Japan. [12]Howard Hughes Medical Institute, Chevy Chase, MD, USA. [13]New York Stem Cell Foundation Robertson Investigator, New York, NY, USA. [14]These authors contributed equally: Daniel J. Emerson, Peiyao A. Zhao, Ashley L. Cook. ✉e-mail: jcremins@seas.upenn.edu

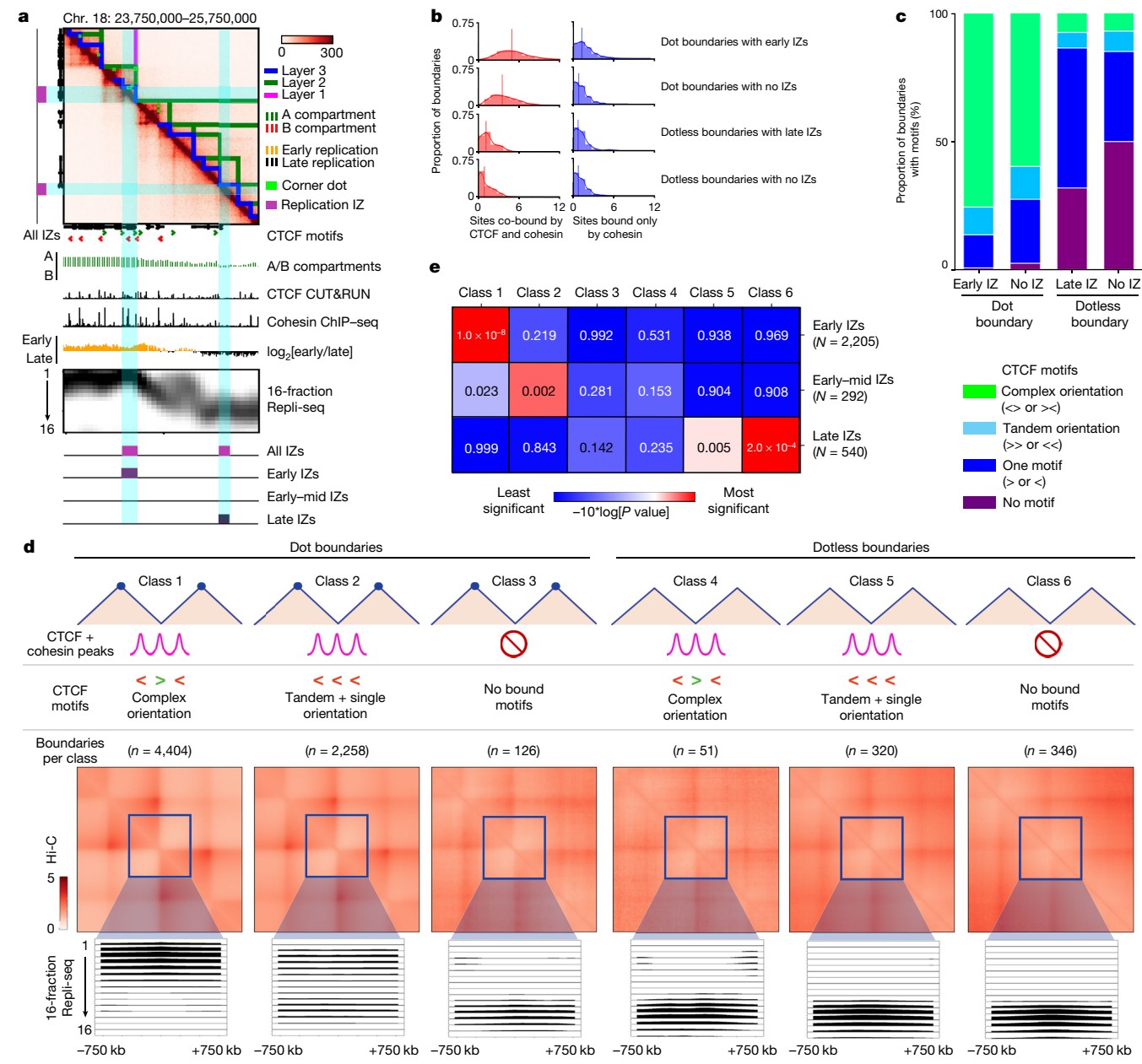

**Fig. 1 | High-efficiency IZs localize specifically to corner-dot TAD/subTAD boundaries with high-density arrays of CTCF + cohesin-binding sites in complex orientations. a**, A Hi-C map from H1 human ES cells for the locus chromosome (chr.) 18: 23.75 Mb–25.75 Mb, hg38, showing TADs, subTADs, loops, CTCF motifs, A/B compartments, CTCF cleavage under targets and release using nuclease (CUT&RUN), cohesin chromatin immunoprecipitation with sequencing (ChIP–seq), two-fraction Repli-seq, 16-fraction Repli-seq and IZs. **b**, Distribution of the number of sites co-bound by CTCF and cohesin (red) or bound only by cohesin (blue) per boundary for: dot boundaries colocalized with early IZs (n = 2,200); dot boundaries colocalized with no IZs (n = 4,087); dotless boundaries colocalized with late IZs (n = 66); and dotless boundaries colocalized with no IZs (n = 628). **c**, Proportion of boundaries with no CTCF motif, one single CTCF motif, CTCF motifs in a tandem orientation, and CTCF motifs in a complex divergent or convergent orientation. Boundaries are stratified into dot and dotless boundaries with either early/late IZs or no IZs. **d**, Top: boundary classification as detailed in the Supplementary Methods and Supplementary Table 7. Middle: aggregate peak analysis of the average observed/expected interaction frequency of the domains centred on each boundary classification. Bottom: averaged 16-fraction Repli-seq signal for each S-phase fraction centred on boundaries ±750 kb. Boundaries, TADs and Repli-seq data were normalized to the same genomic length scale. Boundary numbers are provided only for autosomal chromosomes. **e**, We computed right-tailed, one-tailed empirical P values using a randomization test with early-, early–mid- and late-S-phase IZs and size- and A/B compartment-matched null IZs (Supplementary Methods).

detectable in high-resolution Hi-C data) in the genomic placement of initiated origins following entry into S phase is not known.

We recently developed a high-resolution Repli-seq method to identify the placement of IZs across the genome at 50-kb resolution[16]. We first compared the genomic locations of IZs replicating across early, early–mid and late S phase to our high-resolution Hi-C data developed in the 4D Nucleome Consortium from H1 human embryonic stem (ES) cells[11]. We noticed that high-efficiency, early-S-phase IZs colocalize to strongly insulated boundaries demarcated by corner-dot TADs/sub-TADs on one or both sides (Fig. 1a and Extended Data Figs. 1, 2a and 3).

By contrast, low-efficiency IZs that fire late in S phase can colocalize with boundaries between TADs/subTADs devoid of corner-dots (Fig. 1a and Extended Data Figs. 2b and 3). Our qualitative observations suggest that early and late IZs are enriched at genomic locations serving as boundaries of corner-dot and dotless TAD/subTADs, respectively.

To quantify the link between TAD/subTAD boundaries and IZ genomic placement, we identified a total of 23,851 chromatin domains genome-wide in Hi-C data for human ES cells using our graph-theory-based method 3DNetMod[20] (Supplementary Methods and Supplementary Table 1). We also applied statistical methods developed by our laboratory and others to identify dot-like structures representative of bona fide looping interactions[8,21,22]. We identified 16,922 dots genome wide in ensemble Hi-C maps of human ES cells. Such dots represent punctate groups of adjacent pixels with significantly higher contact frequency compared to the surrounding local chromatin domain structure (Fig. 1a, green rectangles, Supplementary Methods and Supplementary Table 2). After co-registration of dots with domains, we identified 8,279 corner-dot TADs/subTADs and 15,572 dotless TADs/subTADs genome wide in human ES cells (Supplementary Table 3). We stratified boundaries into three groups, including those that are structurally demarcated by: adjacent corner-dot TADs/subTADs on both sides (double-dot boundaries, $n = 6{,}318$); corner-dot TADs/subTADs on only one side and dotless on the other (single-dot boundaries, $n = 2{,}163$); and adjacent dotless TADs/subTADs on both sides (dotless boundaries, $n = 1{,}089$) (Supplementary Table 4). By applying a range of parameter stringencies and methods for dot calling, we could modify the proportion of boundaries classified as double-dot, single-dot and dotless, but the colocalization of dot boundaries with IZs was evident regardless of statistical methodology (Supplementary Methods and Extended Data Fig. 4). We combined all double-dot and single-dot boundaries into dot boundaries, as they showed similar IZ localization patterns (Supplementary Table 4).

Cohesin is essential for the formation of TADs/subTADs through loop extrusion and stalling against boundaries insulated by the architectural protein CTCF[12,13,23–25]. We reasoned that the density and orientation of CTCF-binding sites might reveal an architectural protein signature at boundaries linked to placement of active origins that fire in S phase. We observed a substantially higher density of co-bound CTCF + cohesin-binding sites at dot boundaries overlapping early IZs compared to those that do not overlap any IZs (Fig. 1b and Supplementary Tables 5 and 6). We also examined sites that bind only cohesin, as they can earmark CTCF-independent enhancer–promoter interactions[7,23], but we did not see a notable difference in the number of sites that bind only cohesin across dot versus dotless TAD/subTAD boundaries (Fig. 1b). Together, our data indicate that boundaries colocalizing with human early-S-phase IZs exhibit enriched occupancy of motifs co-bound by CTCF and cohesin, but not cohesin alone, thus confirming and substantially expanding on observations in previous reports linking cohesin generally to a small subset of replication origins in *Drosophila*[26] and humans[27].

Recent reports have uncovered that convergently oriented CTCF motifs anchor long-range looping interactions formed by cohesin-mediated extrusion[12,14,23,28,29]. We observed that most dot boundaries are marked by two or more CTCF + cohesin-bound motifs arranged in a convergent or divergent orientation (hereafter called complex motif orientation; Fig. 1c), and this molecular signature was further enriched when dot boundaries colocalize with early replicating IZs. By contrast, nearly all dotless boundaries have only one or no CTCF + cohesin-bound motifs (Fig. 1c). Dotless boundaries colocalized with late IZs were most often anchored by one CTCF motif. We therefore establish six boundary classes by stratifying dot (classes 1–3) and dotless (classes 4–6) boundaries into those localized with CTCF + cohesin-bound motifs in a complex orientation (classes 1 and 4), tandem or single-motif orientation (classes 2 and 5), or no bound motifs (classes 3 and 6; Fig. 1d).

We next formulated a statistical test to quantify IZ enrichment at boundaries compared to the background expectation across autosomes (Supplementary Methods and Supplementary Table 7). Consistent with our qualitative observations, high-efficiency IZs firing in early S phase were significantly enriched at dot boundaries marked by CTCF + cohesin-binding sites in complex orientations compared to a null distribution of random intervals matched by size and A/B compartment distribution (class 1; Fig. 1d,e, Extended Data Fig. 5b–d and Supplementary Methods). By contrast, low-efficiency IZs firing in late S phase were depleted at dot boundaries and significantly enriched at dotless boundaries with tandem + single CTCF + cohesin-bound motifs or no bound motifs (classes 5 and 6; Fig. 1d,e, Extended Data Fig. 5b–d and Supplementary Methods). We note that our null distribution was created with random intervals matched to real IZs by their size and compartment distribution, reinforcing that the enrichment reflects a strong localization at boundaries above the known link between early and late replication and A and B compartments, respectively (Supplementary Methods).

We sought to independently verify our observed link between IZs and boundaries with an orthogonal technique for assaying replication origin activity. Small nascent strand sequencing (SNS-seq) identifies approximately 10 origins per 100 kb of the genome and enriches for high-efficiency origins localized in early replicating regions[30]. A previous report using ENCODE (Encyclopedia of DNA Elements) phase I pilot microarray data of 1% of the human genome reported enrichment of the cohesin subunit RAD21 at approximately 300 replication origins[27]. Here, using genome folding features from high-resolution Hi-C data, we find that SNS-seq data from human ES cells[30] exhibits heightened origin enrichment specifically at class 1 dot boundaries (Extended Data Fig. 5e). Thus, through two independent replication mapping techniques, we observe a strong enrichment of high-efficiency, early-S-phase IZs at a subset of genetically encoded corner-dot TAD/subTAD boundaries. The colocalization of IZs with TAD boundaries generally has been further confirmed recently with super-resolution imaging[31].

Transcription correlates with origin placement and efficiency[15,17,32–35]. To ascertain whether transcription at boundaries could explain our results, we stratified dot boundaries with a complex CTCF orientation (class 1), dotless boundaries with a complex CTCF orientation (class 4) and dotless boundaries with no CTCF occupancy (class 6) into those that also had transcribed genes and those that were devoid of genes or had only inactive genes (Extended Data Fig. 6 and Supplementary Table 8). Boundaries with transcribed genes in the absence of the dot features (Extended Data Fig. 6b) or in the absence of CTCF + cohesin (Extended Data Fig. 6c) did not exhibit precise localization of high-efficiency early IZs. These results are consistent with the literature, as a large proportion of active promoters are not sites of efficient replication initiation, suggesting that further distinguishing features encode human origins[36]. It is also particularly noteworthy that we see enrichment of early IZs at dot boundaries with a complex CTCF motif orientation only when transcribed genes were also present (Extended Data Fig. 6a). Our data suggest that transcription alone is not sufficient to localize high-efficiency early IZs at boundaries. Transcription may cooperate with CTCF and cohesin-based loop extrusion to position high-efficiency IZs replicating in early S phase.

To understand whether cohesin and TAD/subTAD structural integrity are functionally necessary for origin placement in S phase, we examined IZs after global genome folding disruption using wild-type HCT116 cells engineered to degrade the cohesin subunit RAD21 within hours using a degron[23]. Such a system is uniquely suited to test the role of cohesin-mediated extrusion on IZs decoupled from transcription, as only hours of RAD21 degradation results in genome-wide ablation of nearly all loops with minimal short-term effect on transcription[23]. We synchronized HCT116 RAD21–mAID cells in mitosis, degraded RAD21 with auxin throughout G1, and then assessed replication initiation

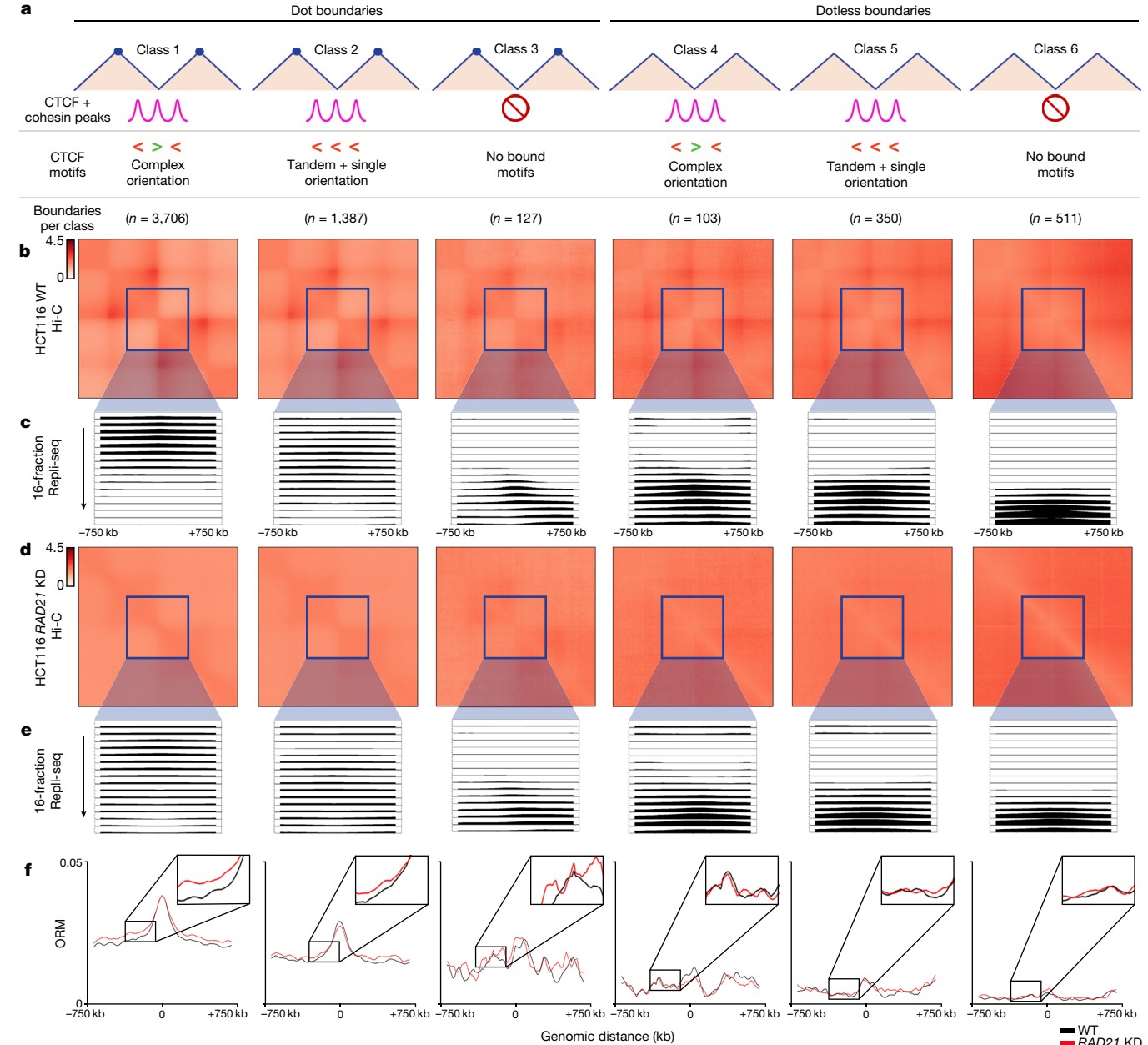

**Fig. 2 | Loss of cohesin-mediated TADs/subTADs severely disrupts the genomic placement of DNA replication IZs. a**, Boundary classification in HCT116 wild-type (untreated HCT116 RAD21–mAID) cells conducted as in human ES cells (Fig. 1d) with boundary counts as listed in Supplementary Table 14. The boundary numbers in the figure are provided for autosomal chromosomes alone. **b**,**d**, Aggregate peak analysis of the Hi-C observed/ expected average interaction frequency of the domains centred on each boundary classification in HCT116 wild-type (WT; untreated HCT116 RAD21–mAID; **b**) and HCT116 *RAD21*-knockdown (KD; auxin-treated HCT116 RAD21–mAID; **d**) cells after cohesin degradation with auxin treatment. The Hi-C source data are from ref. [23]. **c**,**e**, High-resolution 16-fraction Repli-seq data in wild-type HCT116 (WT; untreated HCT116 RAD21–mAID; **c**) and HCT116 *RAD21*-knockdown (KD; auxin-treated HCT116 RAD21–mAID; **e**) cells. Each row represents a temporal fraction from S phase, with 16 rows/fractions in total. The Repli-seq signal plotted represents an average across all boundaries in a particular class for that fraction (*y*-axis) in 50-kb bins across a ±750-kb genomic distance centred on the midpoint of the boundaries (*x*-axis). Sample sizes for each class are shown in **a**. **f**, ORM data for wild-type (untreated HCT116 RAD21–mAID; black) and *RAD21*-knockdown (auxin-treated HCT116 RAD21–mAID; red) cells.

across S phase (Extended Data Fig. 7 and Supplementary Methods). We identified the same dot and dotless TADs/subTADs and boundary classes in Hi-C from wild-type HCT116 (untreated HCT116 RAD21–mAID) cells as in human ES cells (Fig. 2a and Supplementary Tables 9–14). Consistent with previous reports[23], our observations show that nearly all dot and dotless boundaries were destroyed following short-term cohesin knockdown in HCT116 cells (Fig. 2b,d and Extended Data Fig. 8). Therefore, although the molecular composition of boundaries influences their structural features of insulation strength and corner-dot presence, most are dependent on cohesin.

Previous studies have reported that replication timing domains are not globally altered following genome-wide disruption of cohesin-mediated loops[37–39]. Analyses in these studies relied on the log ratio of DNA synthesized in the first or second halves of S phase (two-fraction early/late Repli-seq)[40], the resolution of which renders it difficult to discern IZs. Moreover, previously published two-fraction

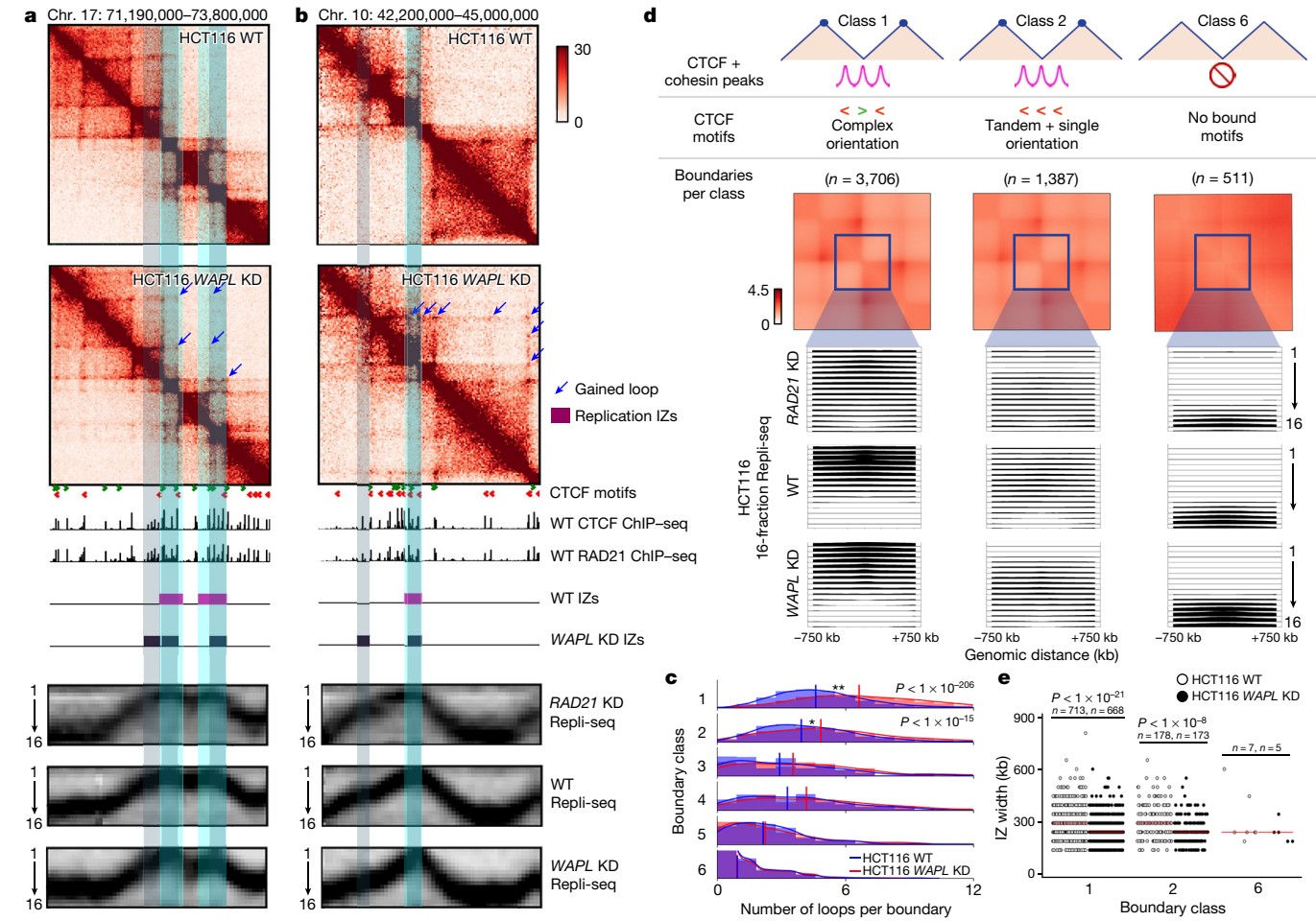

**Fig. 3 | Gain of looping with WAPL degradation narrows the genomic placement of early IZs at dot boundaries with a complex CTCF motif orientation. a,b**, Hi-C maps from wild-type HCT116 (WT; untreated HCT116 WAPL–mAID2) and HCT116 WAPL-knockdown (KD; auxin-treated HCT116 WAPL–mAID2) cells for the loci chromosome 17: 71.19–73.8 Mb, hg38 (**a**) and chromosome 10: 42.2–45 Mb, hg38 (**b**). The tracks show CTCF motifs, CTCF ChIP–seq, RAD21 ChIP–seq, high-resolution 16-fraction Repli-seq and IZs. **c**, Distribution of loops per boundary for each of the six boundary classes. Vertical lines demarcate mean number of loops per boundary within each sample and boundary class. Two-tailed Mann–Whitney $U$-test between HCT116 WAPL-knockdown and HCT116 wild-type cells for class 1 $P = 2.0 \times 10^{-207}$ and class 2 $P = 1.2 \times 10^{-16}$. **d**, Averaged Repli-seq for each of the 16 fractions in a ±750-kb window at boundary classes 1, 2 and 6 as detailed in the Supplementary Methods and Supplementary Table 14. Boundary numbers are provided in the figure for autosomal chromosomes alone. Each Repli-seq row represents a temporal fraction from S phase, there are 16 rows/fractions, and the Repli-seq signal plotted represents an average across all boundaries in a particular class for that fraction (y-axis) in 50-kb bins across a ±750-kb genomic distance centred on the midpoint of boundaries (x-axis). **e**, Width of all IZs colocalized with boundary classes 1, 2 and 6. Two-tailed Mann–Whitney $U$ comparing HCT116 wild-type to HCT116 WAPL-knockdown cells for class 1 $P = 3.0 \times 10^{-22}$ and class 2 $P = 3.3 \times 10^{-9}$.

Repli-seq signals were often quantile normalized[37,39], which obscures the localized disruption in IZ placement and timing shifts at specific TAD/subTAD boundaries. We generated and analysed high-resolution 16-fraction Repli-seq data (Fig. 2c,e and Supplementary Table 15), as well as single-molecule optical replication mapping (ORM) data[17] (Fig. 2f), in both wild-type and cohesin-knockdown HCT116 cells (Extended Data Fig. 7 and Supplementary Methods). As in human ES cells, we observed that 16-fraction Repli-seq data exhibit focal enrichment of high-efficiency/early IZs specifically at dot boundaries marked by CTCF + cohesin co-bound motifs in a complex orientation in wild-type HCT116 cells (class 1; Fig. 2c). Enrichment of early IZs occurs only at boundaries that colocalize with cohesin (Extended Data Fig. 9). Moreover, as in human ES cells, low-efficiency, late IZs were enriched at weak dotless boundaries in wild-type HCT116 cells (Fig. 2c). Using single-molecule ORM data, which can directly assess IZ efficiency as the percentage of molecules that initiate within a particular IZ, we detected enriched origin initiation specifically at class 1 boundaries (Fig. 2f).

Together, our single-molecule and ensemble replication initiation data indicate that early-S-phase IZs fire at a key subset of genetically encoded dot boundaries.

Following ablation of cohesin-mediated boundaries (Fig. 2b,d and Extended Data Fig. 8), we observe severe disruption of high-efficiency early-S-phase IZs specifically at class 1 boundaries, as evidenced by a diffuse and delocalized Repli-seq signal (class 1; Fig. 2c,e). Consistent with our qualitative observations, early wave IZs were less numerous and increased in width specifically at dot boundaries with a complex CTCF motif orientation after loss of cohesin (Extended Data Fig. 10 and Supplementary Table 16). We also noticed that low-efficiency IZs shift to replicating at the end of S phase (fractions 14–16) at dotless boundaries following cohesin knockdown (classes 4–6, Fig. 2c,e and Extended Data Fig. 10). Independently conducted ORM analyses confirmed our observations of IZ disruption by cohesin removal (Fig. 2f). Cell cycle progression and 5-bromodeoxyuridine incorporation was not substantially affected by RAD21 knockdown[39] (Extended Data Fig. 7).

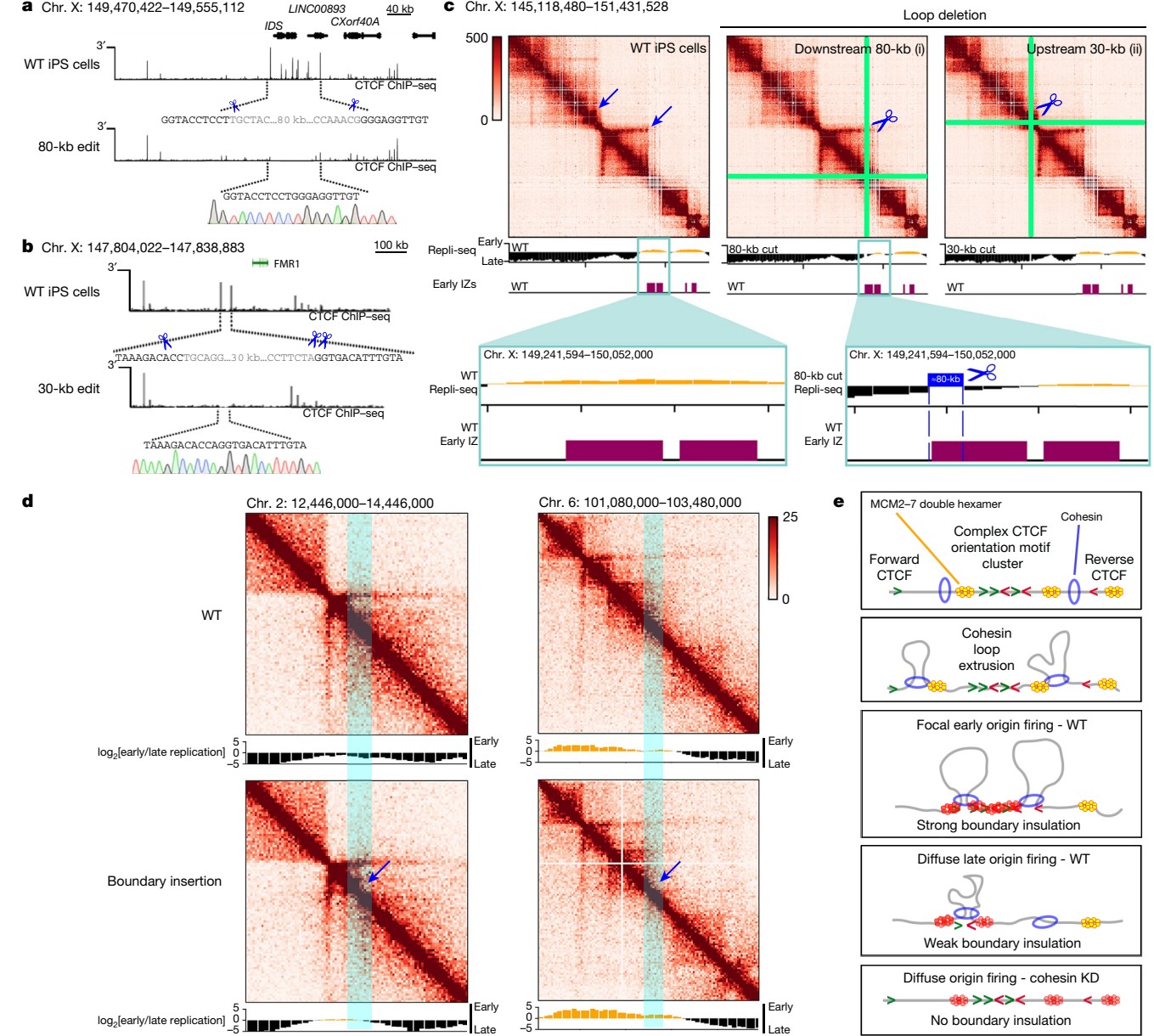

**Fig. 4 | Targeted perturbation leading to gain and loss of structural boundaries can deterministically shift replication timing from early to late S phase. a**, Schematic showing a CRISPR-mediated 80-kb deletion encompassing the *IDS* gene (coordinates of deletion: hg38, chromosome X: 149,470,422–149,555,112). **b**, Schematic showing a CRISPR-mediated 30-kb deletion encompassing two CTCF sites approximately 100 kb upstream from the *FMR1* gene (coordinates of deletion: hg38, chromosome X: 147,804,022–147,838,883). CTCF ChIP–seq tracks for wild-type induced pluripotent stem (iPS) cells and the edited clone are shown. Scissors represent the location of the cut sites verified with Sanger sequencing. **c**, 5C heatmaps (chromosome X: 145,118,480–151,431,528, hg38) and two-fraction Repli-seq tracks in wild-type

iPS cells, and iPS cells with an 80-kb (i) or a 30-kb (ii) loop anchor deletion. Tracks for IZs in human ES cells are overlaid. **d**, Gain of boundary Hi-C and Repli-seq at chromosome 2: ≈13M (hg19) and chromosome 6: ≈102M (hg19) in HAP1 cells with a transposon inserted boundary[43] and replication timing. **e**, Model of DNA replication initiation determined by high-likelihood cohesin extrusion stalling against strong TAD/subTAD boundaries created high-density arrays of CTCF + cohesin-bound motifs with a complex orientation (early replicating IZs) or low-likelihood cohesin pausing against weak TAD/subTAD boundaries formed by single CTCF motifs (late replicating IZs). Yellow double hexamers depict the MCM2–7 complex at licensed origins. Red double hexamers represent the subset of licensed origins that are activated.

Together, our ensemble and single-molecule IZ data demonstrate that disruption of cohesin-mediated loops during G1 alters the genomic placement where origins or clusters of origins fire during early S phase.

On the basis of our observations, we reason that a failure of cohesin to unload, and therefore the creation of new long-range loops due to more cohesin molecules stalled at complex CTCF boundaries in G1 phase, might result in an increased number of high-efficiency origins or a narrowing of their genomic placement in S phase. Recently, it was reported that knockdown of the gene encoding the cohesin unloading

factor WAPL results in increased long-range loops[41]. We examined the genomic placement of IZs in S phase with 16-fraction Repli-seq in wild-type HCT116 cells engineered with an improved degron system (AID2) to degrade WAPL throughout G1 phase[42]. First, we created Hi-C libraries in wild-type and *WAPL*-knockdown HCT116 cells (Fig. 3a,b and Extended Data Fig. 7). Consistent with published results, our observations show that dots indicative of loops are more numerous, and traverse a longer genomic distance, compared with those in wild-type HCT116 cells (Fig. 3a,b and Supplementary Table 18). We observed that

the gain-of-looping phenotype following *WAPL* knockdown occurs most strongly at dot boundaries with a complex CTCF motif orientation (class 1; Fig. 3c). At class 1 boundaries, we observe that early IZs become significantly narrower following *WAPL* knockdown (Fig. 3d,e and Supplementary Table 17). We note that IZs tighten and refine following gain of looping in the *WAPL*-knockdown condition at the same boundaries where IZs grow more diffuse following cohesin knockdown (Fig. 3a,b and Extended Data Fig. 10). Together, the findings from our gain and loss of structural boundary experiments further support a model in which cohesin-based loop extrusion in interphase deterministically informs the placement of the subset of origins that fire during S phase.

We finally sought to understand whether specific boundaries are necessary and sufficient to regulate IZ firing. We used targeted CRISPR–Cas9 genome editing to delete an 80-kb section of the genome containing a complex array of more than 10 CTCF + cohesin-binding sites with complex motif orientations anchoring a long-range chromatin loop that separates late from early replication timing domains (Fig. 4a). The loop anchor was chosen because it also partially overlaps an early-S-phase IZ, but does not encompass the full IZ, thus allowing us to ablate the loop while keeping much of the IZ intact. We observed a striking local delay of replication timing from early to late following deletion of the 80-kb loop anchor, consistent with the loss of an early IZ (Fig. 4a,c(i)). As a negative control, we deleted a different 30-kb loop anchored by two tandemly oriented CTCF-binding sites within an adjacent late replication timing domain, but not overlapping an IZ (Fig. 4b). Deletion of this 30-kb loop anchor disrupted the dot boundary but preserved the timing and genomic location of DNA replication (Fig. 4b,c(ii)). The direct overlap of IZs with boundaries precludes our ability to fully decouple them, and overlap of functional elements remains a technical challenge for functional perturbative studies in the genome biology field at large. Nevertheless, our data provide evidence that replication at a specific early IZ can undergo a striking shift to late S phase following ablation of a boundary. These data are consistent with our cohesin-knockdown observations and our model in which boundaries marked by a complex CTCF motif orientation inform the precise placement of high-efficiency IZs.

As the direct overlap of IZs with boundaries is not amenable to clean, single-variable 'loss-of-structure' perturbative experiments, we also examined a 'gain-of-structure' approach in which we assessed whether the introduction of an engineered ectopic boundary was sufficient to induce changes in replication initiation. We mapped replication with two-fraction Repli-seq in published HAP1 cell lines in which we have previously demonstrated a gain in boundary following insertion of an established 2 kb-sized cell-type-invariant boundary element[43]. We observed a striking shift from late to early replication directly at the location of the engineered boundary (Fig. 4d), consistent with the possibility that boundaries can be sufficient for de novo early IZ firing. Together, our data reveal that both global and local gain and loss of structural boundaries can deterministically influence the placement of IZs.

It is well established that the initiation of DNA replication involves two mutually exclusive steps[1,2]. The first step, origin licensing, begins in telophase with the loading of two copies of the mini-chromosome maintenance (MCM2–7) complex[2,44]. MCM2–7 is initially loaded in excess at tens of thousands of sites across the human genome in an inactive form as a double hexamer that encircles double stranded DNA (yellow double hexamers in Fig. 4e). The second step, origin activation, occurs at the onset of S phase. Origin activation involves mechanisms that both prevent further MCM loading and recruit multiple extra factors to initiate the unwinding of the double helix and DNA synthesis[2,44]. In mammalian systems, a critical mystery remains regarding the mechanisms that governing the selection of a subset of MCM-bound, licensed origins for activation in S phase.

Here we propose a model in which cohesin-mediated loop extrusion and stalling at dot boundaries marked by CTCF + cohesin-binding sites oriented in convergent and divergent directions is required for the positioning of high-efficiency replication origins (Fig. 4e). We propose two possible models to explain the strong localization of high-efficiency IZs to a subset of cohesin-dependent, genetically encoded boundaries: cohesin could directly push licensed MCM double hexamers or other origin activation cofactors along the genome before stalling at high-density arrays of CTCF + cohesin-bound motifs in complex orientations; alternatively, cohesin might pass over many licensed, MCM-bound origins and selectively participate in the activation of those already loaded at boundaries. We also posit that low-efficiency IZs might fire at weaker dotless boundaries later in S phase because cohesin only temporarily pauses during its traversal along the genome, and thus cannot aggregate initiation activity (Fig. 4e). In the cell types from our study, cohesin-mediated loop extrusion is required for IZ placement, and the changes in replication timing are subtle and indirect owing to the altered distance of nearby genomic regions to the nearest initiation site. We note that although we do not see evidence for a dominant role for cohesin on the larger replication timing program, we cannot rule out that cohesin knockdown might have a more profound effect on the replication timing program in other cell types, species and experimental designs.

Previous studies using mass spectrometry and co-immuno-precipitation have reported the direct binding of cohesin to DNA replication factors, such as MCM7, MCM6, MCM4, RFC1 and DNA polymerase α[27,45]. The MCM complex has the ability to slide after loading and can be pushed by polymerase during transcription[46–48]. However, the extent and rate at which this occurs on chromatin in the presence of nucleosomes ($\approx$11 nm) is still an open question. The internal diameter of cohesin is 40 nm, whereas the MCM2–7 double hexamer is only 15 nm. The findings of a recent Hi-C and imaging study suggest that, despite their small size, MCM complexes could also serve as boundaries to block cohesin-based loop extrusion[49]. TAD boundaries and loops persist through S phase[50], but MCMs are removed from chromatin after IZs fire[1,2]. Therefore, we favour a model in which cohesin pushes licensed MCMs in G1, leading to the localization and activation of a key subset of origins at boundaries with a complex CTCF motif orientation in S phase (Fig. 4e). Nevertheless, both proposed models remain exciting areas for future mechanistic dissection.

Understanding the structure–function relationship of the human genome remains a major challenge for human geneticists and chromatin biologists. Here we stratify TADs and subTADs by their structural and molecular features. We conduct global and local perturbative studies to reveal that genetically encoded TAD/subTAD boundaries formed by cohesin-mediated loop extrusion in G1/pre-S functionally inform genome function in the case of the initiation of DNA replication in S phase. Our work sheds light on the question of whether and how the location of fired origins is deterministically encoded in humans by the genome, epigenome and higher-order chromatin folding.

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

**Reporting summary**
Further information on research design is available in the Nature Research Reporting Summary linked to this paper.

## Data availability

All new raw data created in this manuscript have been uploaded to the 4D Nucleome portal and will be freely released for full distribution to the public (see specific details below). Processed data files for all figures and extended data figures are provided as Supplementary Tables 1–19. ORM data have been uploaded to the National Center for Biotechnology Information, BioProject database accession number PRJNA788726 (http://genome.ucsc.edu/s/dsaulebe/ORM%20data%20HCT116). Two-fraction Repli-seq data for Blobel engineered lines (raw data and processed $\log_2$[early/late] from three conditions) were obtained from https://www.ncbi.nlm.nih.gov/geo/query/acc.cgi?acc=GSE190117.

Group 1 data (16-fraction Repli-seq data for H1 human ES cells) are available from the 4D Nucleome portal as follows: H1 human ES raw fastq, https://data.4dnucleome.org/experiment-sets/4DNESXRBILXJ/; H1 human ES read-depth-normalized array for visualization, https://data.4dnucleome.org/files-processed/4DNFIEEYFQ7C/; H1 human ES scaled, read-depth-normalized array for IZ calls, https://data.4dnucleome.org/files-processed/4DNFI3N8GHKR/; H1 human ES early, early–mid and late IZs on read-depth-normalized array, https://data.4dnucleome.org/files-processed/4DNFIRF7WZ3H/.

Group 2 data (16-fraction Repli-seq data for wild-type HCT116 cells) are available from the 4D Nucleome portal as follows: wild-type HCT116 raw fastq, https://data.4dnucleome.org/experiment-sets/4DNESNGZM5FG/; wild-type HCT116 mitochondria-normalized array for IZ calls, https://data.4dnucleome.org/files-processed/4DNFIPIQTMJ9/; wild-type HCT116 early, early–mid and late IZs on mitochondria-normalized array, https://data.4dnucleome.org/files-processed/4DNFI95K53YS/.

Group 3 data (16-fraction Repli-seq data for wild-type and cohesin-knockdown HCT116 pairing) are available from the 4D Nucleome portal as follows: *RAD21*-knockdown HCT116 raw, https://data.4dnucleome.org/experiment-sets/4DNES92AU9JR/; *RAD21*-knockdown HCT116 read-depth-normalized downsampled array for IZ calls, https://data.4dnucleome.org/files-processed/4DNFI3ZMWG5T/; *RAD21*-knockdown HCT116 early, early–mid and late IZs called on the read-depth-normalized downsampled array, https://data.4dnucleome.org/files-processed/4DNFIGOMS9G7/; wild-type HCT116 raw fastq, https://data.4dnucleome.org/experiment-sets/4DNESNGZM5FG/; wild-type HCT116 read-depth-normalized downsampled array for IZ calls, https://data.4dnucleome.org/files-processed/4DNFI6NGWNOG/; wild-type HCT116 early, early–mid and late IZs called on the read-depth-normalized downsampled array, https://data.4dnucleome.org/files-processed/4DNFIYO3H24N/.

Group 4 data (16-fraction Repli-seq data for wild-type and *WAPL*-knockdown HCT116 pairing) are available from the 4D Nucleome portal as follows: *WAPL*-knockdown HCT116 raw, https://data.4dnucleome.org/experiment-sets/4DNES72NE7SL/; *WAPL*-knockdown HCT116 read-depth-normalized downsampled array for IZ calls, https://data.4dnucleome.org/files-processed/4DNFI7MI88QR/; *WAPL*-knockdown HCT116 early, early–mid and late IZs called on the read-depth-normalized downsampled array, https://data.4dnucleome.org/files-processed/4DNFIDI1QJVA/; wild-type HCT116 raw fastq, https://data.4dnucleome.org/experiment-sets/4DNESNGZM5FG/; wild-type HCT116 read-depth-normalized downsampled array for IZ calls, https://data.4dnucleome.org/files-processed/4DNFI6NGWNOG/; wild-type HCT116 early, early–mid and late IZs called on the read-depth-normalized downsampled array, https://data.4dnucleome.org/files-processed/4DNFILNNSFMD/.

Group 5 data (16-fraction Repli-seq data visualization) are available from the 4D Nucleome portal as follows: wild-type HCT116 read-depth-normalized downsampled array for visualization, https://data.4dnucleome.org/files-processed/4DNFI6NGWNOG/; *RAD21*-knockdown HCT116 read-depth-normalized downsampled array for visualization, https://data.4dnucleome.org/files-processed/4DNFI3ZMWG5T/; *WAPL*-knockdown HCT116 read-depth-normalized downsampled array for visualization, https://data.4dnucleome.org/files-processed/4DNFI7MI88QR/.

Hi-C data for wild-type and *WAPL*-knockdown HCT116 pairing are available from the 4D Nucleome portal as follows: *WAPL*-knockdown HCT116 raw Hi-C, https://data.4dnucleome.org/experiment-set-replicates/4DNES1JP4KZ1/; *WAPL*-knockdown HCT116 normalized balanced Hi-C matrices, https://data.4dnucleome.org/files-processed/4DNFIY5939F3/; *WAPL*-knockdown HCT116 loops, https://data.4dnucleome.org/files-processed/4DNFILP7BD5H/; wild-type HCT116 raw Hi-C, https://data.4dnucleome.org/experiment-set-replicates/4DNESNSTBMBY/; wild-type HCT116 normalized balanced Hi-C matrices, https://data.4dnucleome.org/files-processed/4DNFI5MR78O6/; wild-type HCT116 loops, https://data.4dnucleome.org/files-processed/4DNFIOQLL854/.

Two-fraction Repli-seq data for human iPS wild-type and two CRISPR-engineered lines (raw data and processed $\log_2$[early/late] from three conditions) are available from the 4D Nucleome portal as follows: wild-type human iPS line raw data, https://data.4dnucleome.org/experiment-sets/4DNESDYES9QD/; wild-type human iPS line $\log_2$[early/late], https://data.4dnucleome.org/files-processed/4DNFI5WEY784/; human engineered clone 1 80-kb-IZ-deletion iPS line raw data, https://data.4dnucleome.org/experiment-sets/4DNESE3WCUAQ/; human engineered clone 1 80-kb-IZ-deletion iPS line $\log_2$[early/late], https://data.4dnucleome.org/files-processed/4DNFIZMB415V/; human engineered clone 2 30-kb-control-deletion iPS line raw data, https://data.4dnucleome.org/experiment-sets/4DNES66YWJU7/; human engineered clone 2 30-kb-control-deletion iPS line $\log_2$[early/late], https://data.4dnucleome.org/files-processed/4DNFIWDMF7HW/.

5C data for human IPS wild-type and two engineered lines (primer bed file, raw heatmaps and processed heatmaps from three conditions) are available from the 4D Nucleome portal as follows: wild-type human iPS line raw data, https://data.4dnucleome.org/experiment-set-replicates/4DNESLRDUPZ6/; wild-type human iPS line balanced 5C data, replicate 1, https://data.4dnucleome.org/files-processed/4DNFIXM8V3ZB/, replicate 2, https://data.4dnucleome.org/files-processed/4DNFIDB6M1ZN/; wild-type human engineered clone 1 80-kb-boundary-deletion iPS line raw data, https://data.4dnucleome.org/experiment-set-replicates/4DNES39F1QWU/; wild-type human engineered clone 1 80-kb-boundary-deletion iPS line balanced 5C data, https://data.4dnucleome.org/files-processed/4DNFIA8P94BX/; wild-type human engineered clone 2 30-kb-control-deletion iPS line raw data, https://data.4dnucleome.org/experiment-set-replicates/4DNES3PDMUHG/; wild-type human engineered clone 2 30-kb-control-deletion iPS line balanced 5C data: replicate 1, https://data.4dnucleome.org/files-processed/4DNFI7WZYRHP/, replicate 2, https://data.4dnucleome.org/files-processed/4DNFI7V4VXAQ/.

## Code availability

We freely release all custom code for loop, TAD and subTAD detection at the following bitbucket links: TAD/subTAD detection, https://bitbucket.org/creminslab/cremins_lab_tadsubtad_calling_pipeline_11_6_2021; loop detection, https://bitbucket.org/creminslab/cremins_lab_loop_calling_pipeline_11_6_2021/src/initial/.

**Acknowledgements** We thank members of the 4D Nucleome community and the laboratories of J.E.P.-C. and D.M.G. for helpful discussions. J.E.P.-C. is a New York Stem Cell Foundation – Robertson Investigator and an Alfred P. Sloan Foundation Fellow. J.D. is an investigator of the Howard Hughes Medical Institute. This research was supported by National

Institute of Mental Health grants (1R011MH120269 and 1DP1MH129957; J.E.P.-C.), 4D Nucleome Common Fund grants (1U01HL12999801, 1U01DK127405 and 1U01DA052715; J.E.P.-C.), a National Science Foundation (NSF) CAREER Award (CBE-1943945; J.E.P.-C.), an NSF Emerging Frontiers in Research Innovation grant (1933400; J.E.P.-C.), 4D Nucleome Common Fund grants (DK107980 and HG011536; J.D.) and National Institutes of Health grants (R01HG010658 and U54DK107965; D.M.G.). The PhD fellowship to D.S. was provided by the CNRS 80|Prime interdisciplinary programme and W.W. was supported by a COFUND IC-3i International PhD fellowship.

**Author contributions** Conceptualization: D.J.E., P.A.Z., D.M.G., J.E.P.-C. Experimentation: C.G., L.Z., Z.S., K.K., W.G., L.Y., J.H.G., T.S., H.Y., F.Y., J.D., D.M.G., J.E.P.-C. Computation and visualization: D.J.E., P.A.Z., A.L.C., M.K.M., R.J.B., K.R.T., S.V.V., D.S., W.W., C.-L.C., D.M.G., J.E.P.-C. Funding acquisition: J.D., D.M.G., J.E.P.-C. Project administration: D.M.G., J.E.P.-C. Writing: D.J.E., P.A.Z., A.L.C., J.E.P.-C. Review and editing: D.J.E., P.A.Z., A.L.C., M.K.M., J.D., C.-L.C., D.M.G., J.E.P.-C. Critical reagents: D.Z., M.T.K.

**Competing interests** The authors declare no competing interests.

**Additional information**
**Correspondence and requests for materials** should be addressed to Jennifer E. Phillips-Cremins.

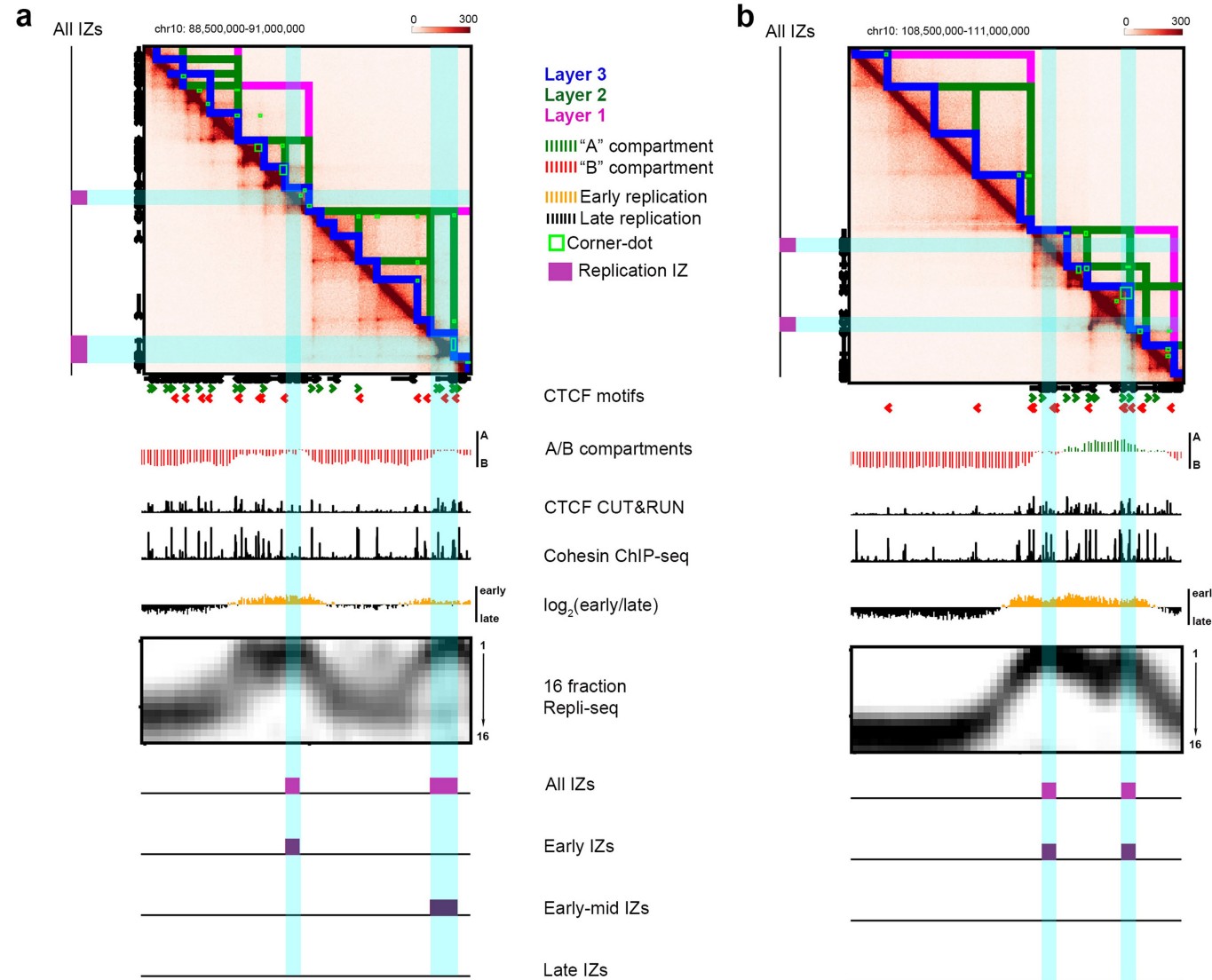

**Extended Data Fig. 1 | Hi-C and 16-fraction Repli-seq in H1 human ES cells.**
Hi-C maps showing **(a)** chr10: 88.5 - 91.0 Mb locus and **(b)** chr10: 108 - 111 Mb locus. Blue lines, Layer 3 most nested TADs/subTADs. Green lines, Layer 2 intermediate TAD/subTAD nesting. Magenta lines, Layer 1 highest-layer non-nested TADs/subTADs. Green rectangles, corner-dots. Tracks show CTCF motifs at colocalized CTCF+cohesin peaks (green, forward; red, reverse), A/B compartments (green, A compartment; red, B compartment), CTCF CUT&RUN and cohesin ChIP-seq (black), low-resolution two-fraction replication timing domains (yellow, early replication timing; black, late replication timing), 16-fraction Repli-seq data, and initiation zones (magenta, all IZs; purple, Early/Early-mid/Late IZs). Genome build, hg38.

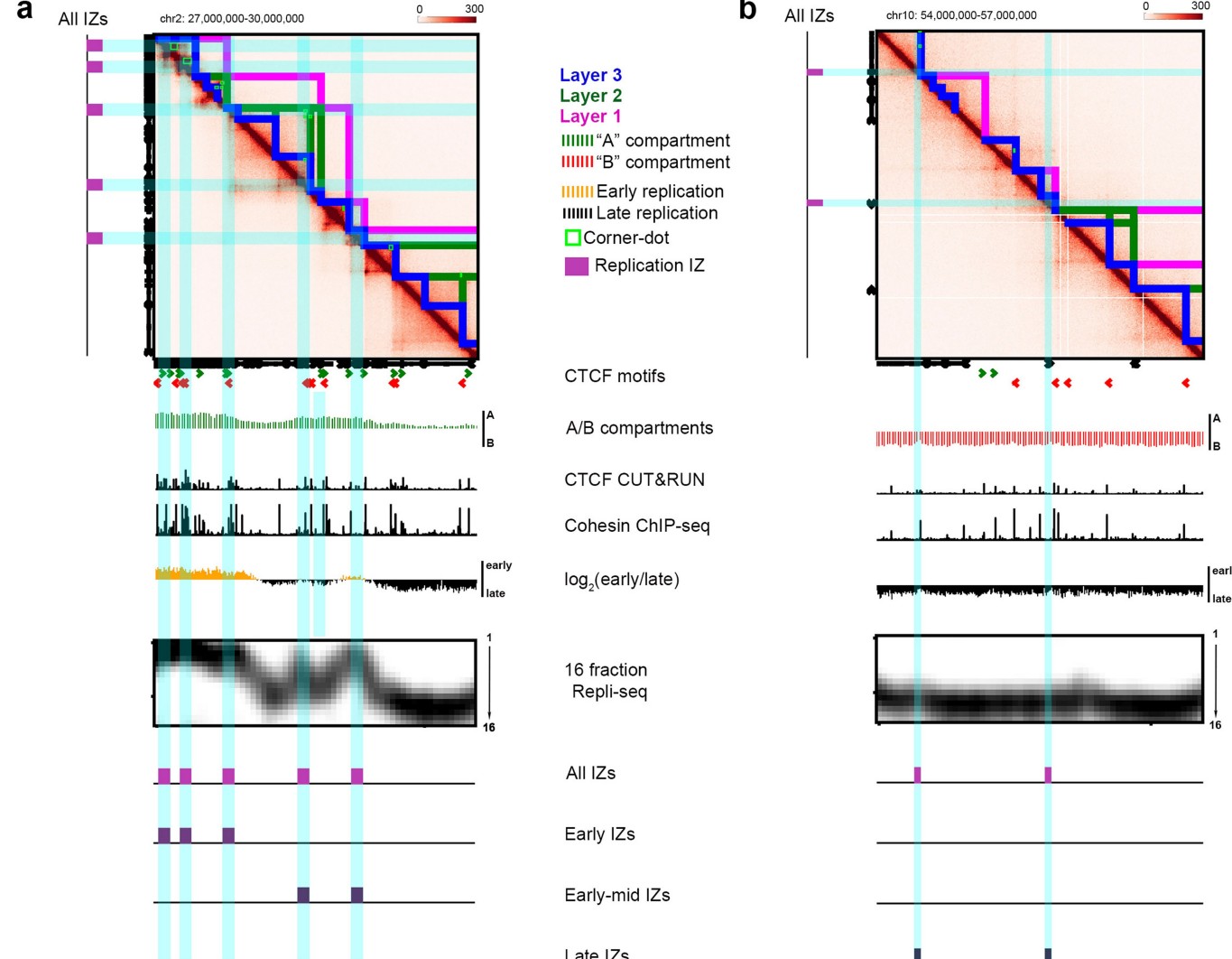

**Extended Data Fig. 2 | Hi-C and 16-fraction Repli-seq in H1 human ES cells.**
Hi-C maps showing (a) chr2: 27 - 30 Mb locus and (b) chr10: 54 - 57 Mb locus. Blue lines, Layer 3 most nested TADs/subTADs. Green lines, Layer 2 intermediate TAD/subTAD nesting. Magenta lines, Layer 1 highest-layer non-nested TADs/subTADs. Green rectangles, corner-dots. Tracks show CTCF motifs at colocalized CTCF+cohesin peaks (green, forward; red, reverse), A/B compartments (green, A compartment; red, B compartment), CTCF CUT&RUN and cohesin ChIP-seq (black), low-resolution two-fraction replication timing domains (yellow, early replication timing; black, late replication timing), 16-fraction Repli-seq data, and initiation zones (magenta, all IZs; purple, Early/Early-mid/Late IZs). Genome build, hg38.

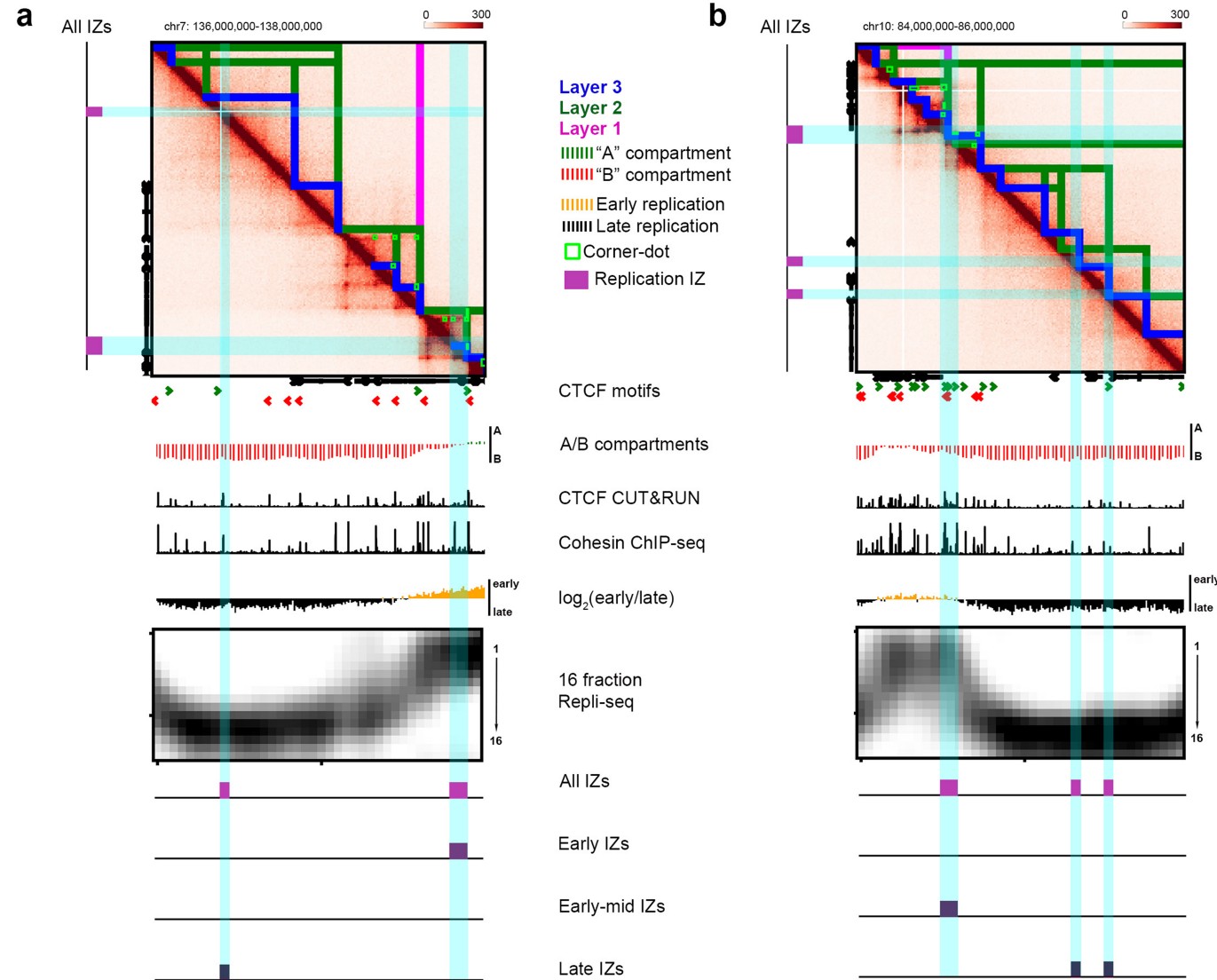

**Extended Data Fig. 3 | Hi-C and 16-fraction Repli-seq in H1 human ES cells.** Hi-C maps showing **(a)** chr7: 136 – 138 Mb locus and **(b)** chr10: 84 – 86 Mb locus. Blue lines, Layer 3 most nested TADs/subTADs. Green lines, Layer 2 intermediate TAD/subTAD nesting. Magenta lines, Layer 1 highest-layer non-nested TADs/subTADs. Green rectangles, corner-dots. Tracks show CTCF motifs at colocalized CTCF+cohesin peaks (green, forward; red, reverse), A/B compartments (green, A compartment; red, B compartment), CTCF CUT&RUN and cohesin ChIP-seq (black), low-resolution two-fraction replication timing domains (yellow, early replication timing; black, late replication timing), 16-fraction Repli-seq data, and initiation zones (magenta, all IZs; purple, Early/Early-mid/Late IZs). Genome build, hg38.

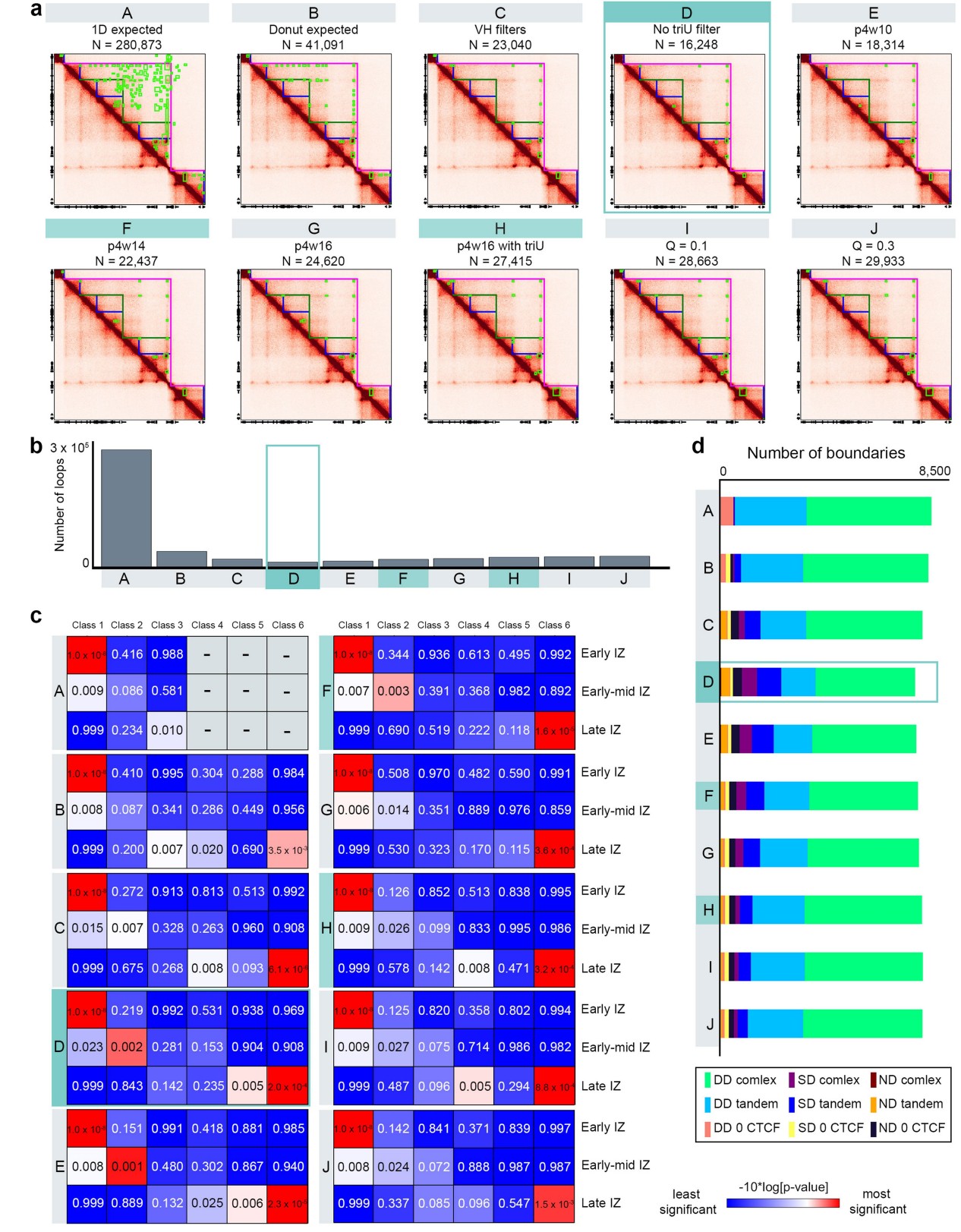

**Extended Data Fig. 4** | See next page for caption.

**Extended Data Fig. 4 | Loop calling in H1 human ES Hi-C across a series of single-variable parameter changes. (a)** Example genomic locus in human ES cells (chr4: 52.9 - 54.9 Mb, hg38, scale 0–200) with 10 methodological variants of corner-dot detection (Options A through J detailed in the Supplementary Methods). In teal, we highlight Options D, F, and H as our recommended loop calling parameters in Hi-C 2.5 generated from human ES cells for conservative, intermediate, and permissive calls respectively. Option D – our conservative loop calling set – is indicated by a teal box and was used to call loops for the analysis in the main paper. **(b)** Bar graph showing the number of loops called across autosomes for loop calling parameters Options A through J. **(c)** We computed right-tailed, one-tailed empirical p-values using a randomization test with Early, Early-mid, and Late S phase IZs and size- and A/B compartment-matched null IZs (Supplemental Methods) across boundaries derived from Options A-J dot calling variants. **(d)** Number of TAD/subTAD boundaries from autosomal chromosomes classified into the following categories: double-dot complex CTCF motif orientation (DD complex), double-dot tandem + single CTCF motif orientation (DD tandem), double-dot no CTCF (DD 0 CTCF), single-dot CTCF complex motif orientation (SD complex), single-dot CTCF tandem + single motif orientation (SD tandem), single-dot no CTCF (SD 0 CTCF), dotless complex CTCF motif orientation (ND complex), dotless tandem + single CTCF motif orientation (ND tandem), and dotless no CTCF (ND 0 CTCF).

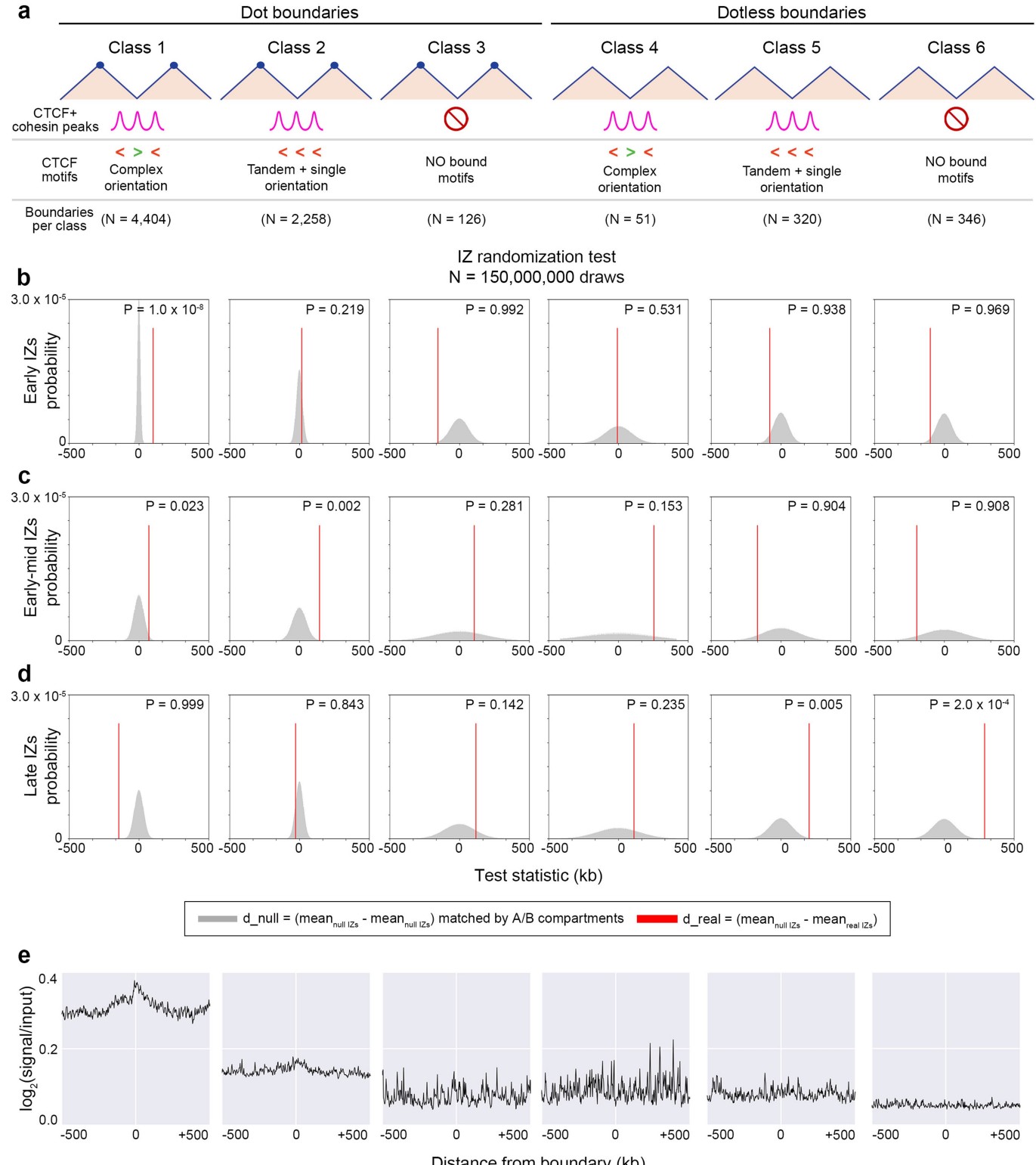

**Extended Data Fig. 5** | See next page for caption.

**Extended Data Fig. 5 | Statistical test and SNS-seq in H1 human ES cells reveals the enrichment of Early IZs at class 1 boundaries. (a)** Boundary classification schematic in human ES cells with the following boundary counts: (i) N = 4,404, (ii) N = 2,258, (iii) N = 126, (iv) N = 51, (v) N = 320, (vi) N = 346. Boundary class numbers in figure and caption provided for autosomal chromosomes only. **(b-d)** Statistical test computing proximity of IZs to TAD/subTAD boundaries compared to expectation in hES Hi-C autosomes. We computed right-tailed, one-tailed empirical p-values using a randomization test with **(b)** early, **(c)** early-mid, and **(d)** late S phase IZs and size- and A/B compartment-matched null IZs (Supplemental Methods). Test statistic for real IZs (red line) represents the difference between the average null IZ distance to closest boundary and average real IZ distance to closest boundary (detailed in Supplemental Methods). Null distribution represents the difference between the average distance to the closest boundary of two reshuffled sets of null IZs. **(e)** We plotted the average SNS-seq signal (reads per million) 500 kb up- and down-stream of the 6 boundary classes. SNS-seq data in human ES cells was acquired from Besnard et al[51]. The overall level of SNS-seq signal at Dot boundaries was also higher than Dotless boundaries, reinforcing the shared propensity of SNS-seq origins and corner-dot TADs/subTADs to both be enriched in the same genomic compartment (A compartment), which we controlled for in our statistical tests.

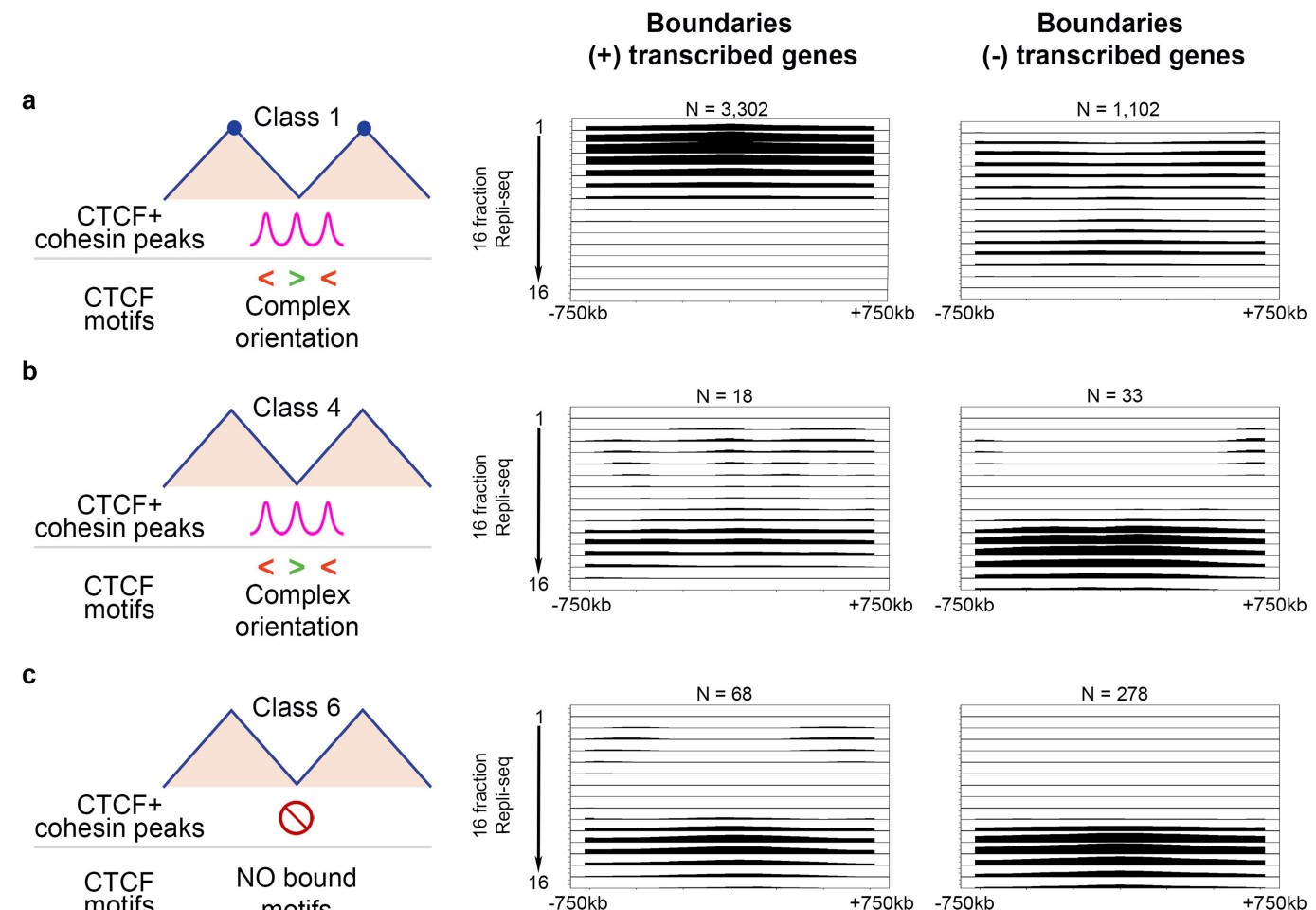

**Extended Data Fig. 6 | Patterns of 16-fraction Repli-seq in boundaries +/− transcribed genes in H1 human ES cells.** Repli-seq was averaged for each of the 16 fractions in a +/− 750 kb window at **(a)** Boundary class 1, dot boundaries with complex CTCF motif orientation, **(b)** Boundary class 4, dotless boundaries with complex CTCF motif orientation, and **(c)** Boundary class 6, dotless boundaries with no CTCF, further stratified by colocalization with transcribed genes (+ transcription) vs. no genes & no transcribed genes (- transcription) within +/−100 kb of the midpoint of the boundary. 16 Fraction Repli-seq images pileup scale 0.6–1.85. Boundary numbers provided in figure for autosomal chromosomes only.

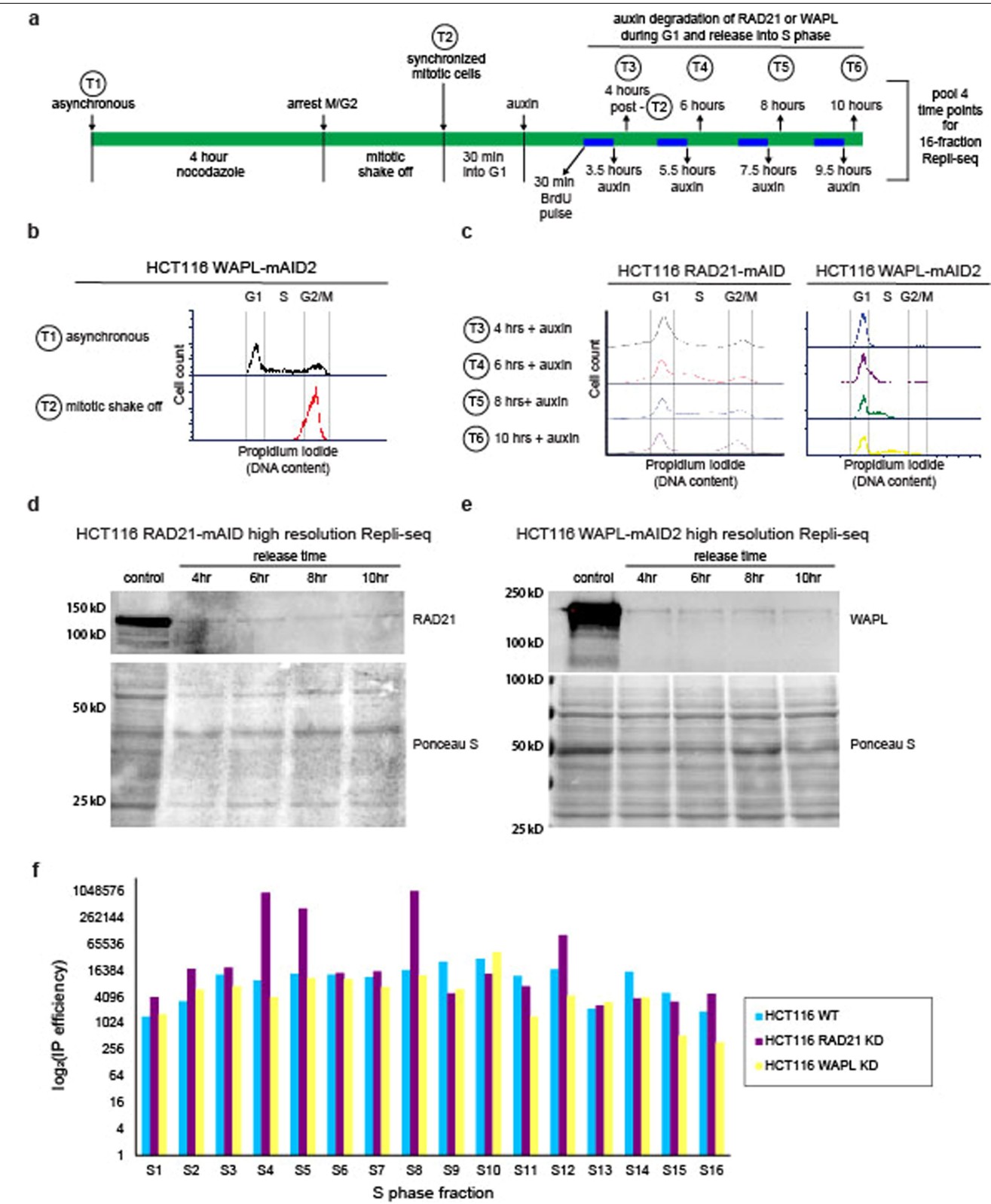

**Extended Data Fig. 7** | See next page for caption.

**Extended Data Fig. 7 | HCT116 characterization leading to the generation of wild type, WAPL knock-down, and RAD21 knock-down genomics libraries. (a)** Treatment and sample collection timeline of HCT116 RAD21-mAID and HCT116 WAPL-mAID2 cells for high-resolution 16-fraction Repli-seq. **(b-c)** Propidium Iodide FACS histograms measuring DNA content for **(b)** HCT116 WAPL-mAID2 cells in asynchronous cultures and immediately after mitotic shake-off conditions, **(c)** auxin-treated HCT116 RAD21-mAID cells and HCT116 WAPL-mAID2 cells at specified time points after mitotic shake-off. No clear defect in cell cycle progression was observed. **(d)** Western blot of RAD21 protein in HCT116 RAD21-mAID cells for untreated control and timepoints after auxin treatment post mitotic shake off. Ponceau S stain for total protein. Blot run on one set of samples. **(e)** Western blot of WAPL protein in HCT116 WAPL-mAID2 cells for untreated control and timepoints after auxin treatment post mitotic shake off. Ponceau S stain for total protein. Blot run on one set of samples. **(f)** Total IP efficiency (genomic DNA over mitochondrial DNA) for each of 16 S phase fractions of high-resolution 16-fraction Repli-seq for HCT116 WT, HCT116 RAD21-mAID KD, and HCT116 WAPL-mAID2 KD cells.

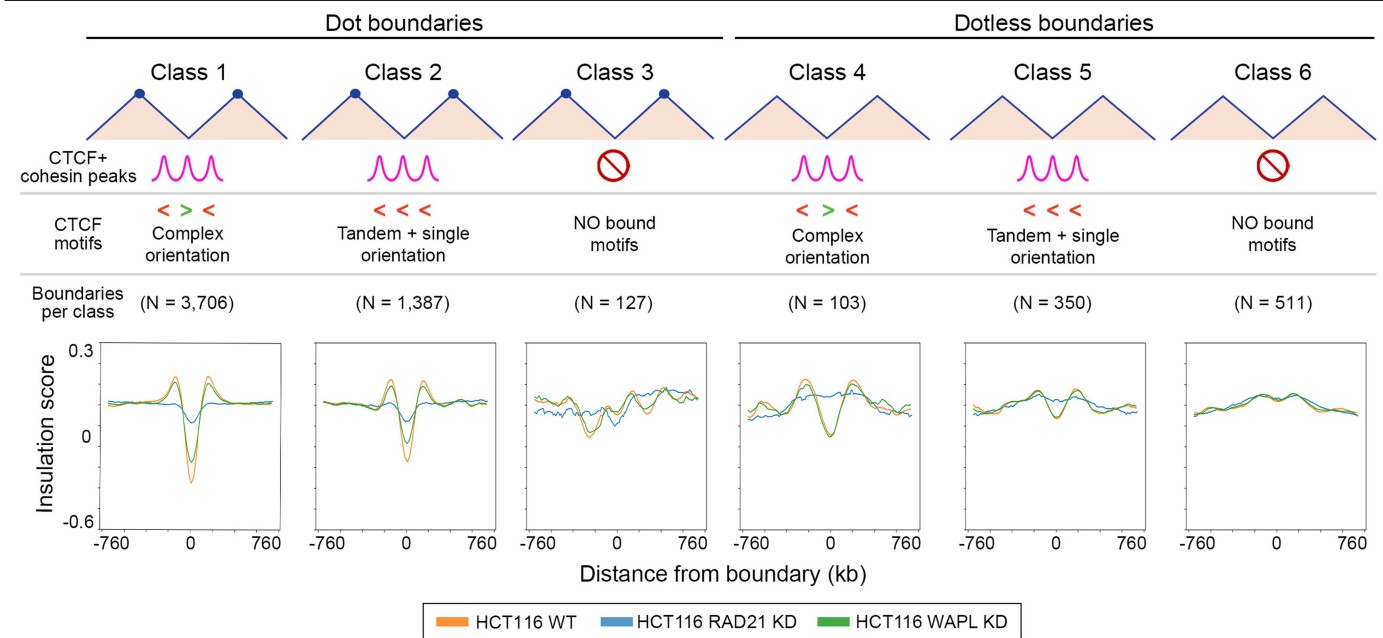

**Extended Data Fig. 8 | Insulation score changes at Boundary classes 1–6 in HCT116 cells upon cohesin and WAPL knock-down.** Average insulation score for each of the six boundary classes in a +/− 760 kb window for wild type HCT116 (WT; untreated HCT116 WAPL-mAID2), HCT116 with cohesin knock-down (RAD21 KD; auxin-treated HCT116 RAD21-mAID), and HCT116 with WAPL knock-down (WAPL KD; auxin-treated HCT116 WAPL-mAID2). The six boundary classes have the following boundary counts (i) N = 3,706 (ii) N = 1,387, (iii) N = 127, (iv) N = 103, (v) N = 350, (vi) N = 511. Boundary numbers provided in figure and caption for autosomal chromosomes only.

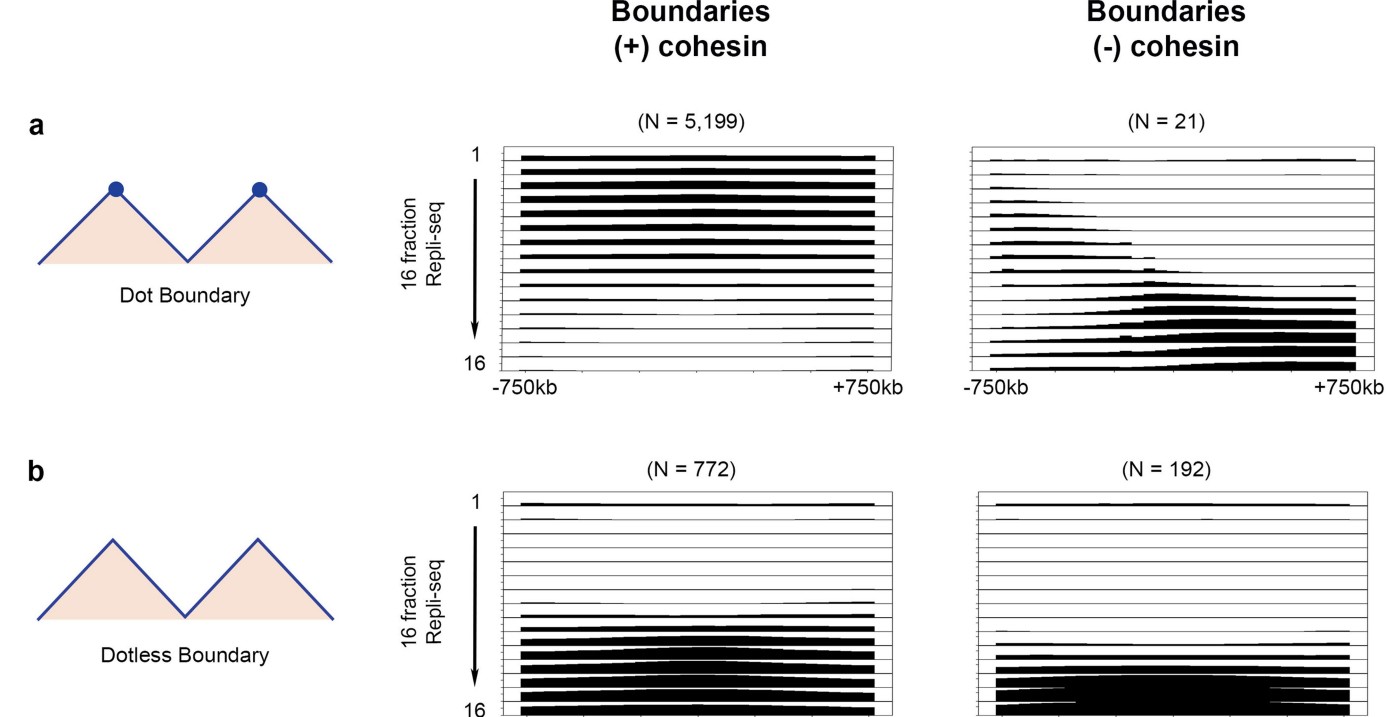

**Extended Data Fig. 9 | Patterns of 16-fraction Repli-seq in Dot versus Dotless boundaries +/- cohesin in HCT116.** Repli-seq was averaged for each of the 16 fractions in a +/− 750 kb window at **(a)** all boundaries with demarcating dot TAD/subTADs on one or both sides, Boundary classes 1–3, or **(b)** all boundaries with dotless TADs/subTADs on both sides, Boundary classes 4–6, further stratified by the colocalization with cohesin ChIP-seq peaks (+ cohesin) vs. no cohesin peaks (- cohesin) within +/− 100 kb of the boundary. 16 Fraction Repli-seq images with cohesin pileup scale 5.0–9.8, no cohesin pileup scale 5.0–13.0. Boundary numbers provided in figure for autosomal chromosomes only.

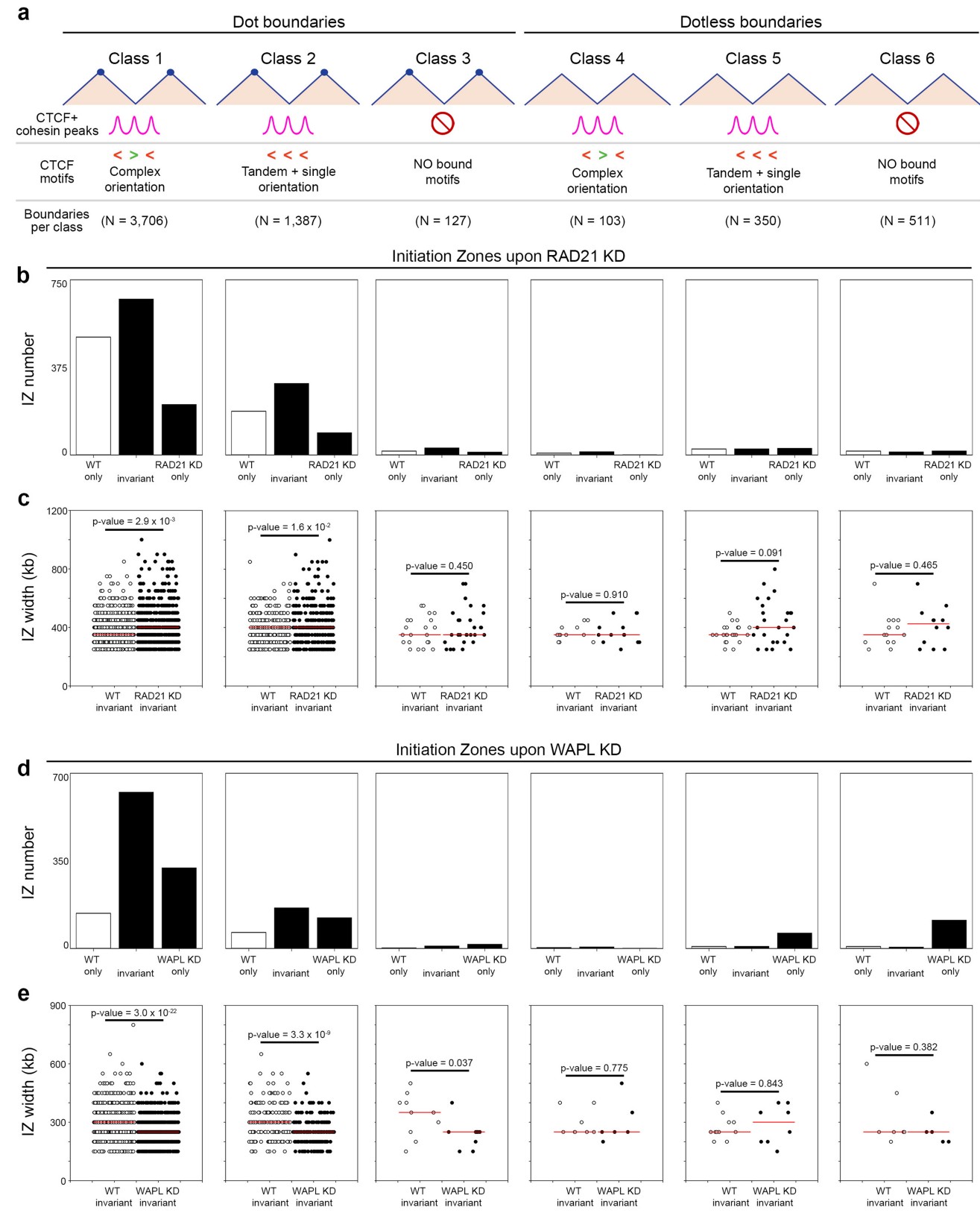

**Extended Data Fig. 10 |** See next page for caption.

**Extended Data Fig. 10 | Number and width of IZs in HCT116 cells upon cohesin and WAPL knock-down. (a)** Boundary classes in HCT116 with the following boundary counts (i) N = 3,706 (ii) N = 1,387, (iii) N = 127, (iv) N = 103, (v) N = 350, (vi) N = 511. Boundary numbers provided in figure and caption for autosomal chromosomes only. **(b)** Number of IZs in WT only (unique to wild type HCT116 – white bar), invariant (wild type HCT116 overlapping auxin-treated HCT116 RAD21-mAID KD – middle black bar), and RAD21 KD only (unique to auxin-treated HCT116 RAD21-mAID KD – right black bar). **(c)** Width of all IZs colocalized with Boundary classes 1 - 6 in HCT116 WT and HCT116 RAD21 KD conditions. Median value indicated by red line. Two-tailed Mann-Whitney U comparing overlapping IZs in HCT116 WT and HCT116 RAD21 KD samples. Only IZs overlapping in both HCT116 wild type and HCT116 RAD21 knock-down are plotted. **(d)** Number of IZs in WT only (unique to wild type HCT116 – white bar), invariant (wild type HCT116 overlapping auxin-treated HCT116 WAPL-mAID2 KD – middle black bar), and WAPL KD only (unique to auxin-treated HCT116 WAPL-mAID2 KD – right black bar). **(e)** Width of all IZs colocalized with Boundary classes 1 - 6 in HCT116 WT and HCT116 WAPL KD conditions. Median value indicated by red line. Two-tailed Mann-Whitney U comparing overlapping IZs in HCT116 WT and HCT116 WAPL KD samples. Only IZs overlapping in both HCT116 wild type and HCT116 WAPL knock-down are plotted.

# nature research

# Reporting Summary

Nature Research wishes to improve the reproducibility of the work that we publish. This form provides structure for consistency and transparency in reporting. For further information on Nature Research policies, see our Editorial Policies and the Editorial Policy Checklist.

## Statistics

For all statistical analyses, confirm that the following items are present in the figure legend, table legend, main text, or Methods section.

| n/a | Confirmed | |
|---|---|---|
| ☐ | ☒ | The exact sample size (*n*) for each experimental group/condition, given as a discrete number and unit of measurement |
| ☐ | ☒ | A statement on whether measurements were taken from distinct samples or whether the same sample was measured repeatedly |
| ☐ | ☒ | The statistical test(s) used AND whether they are one- or two-sided<br>*Only common tests should be described solely by name; describe more complex techniques in the Methods section.* |
| ☒ | ☐ | A description of all covariates tested |
| ☐ | ☒ | A description of any assumptions or corrections, such as tests of normality and adjustment for multiple comparisons |
| ☒ | ☐ | A full description of the statistical parameters including central tendency (e.g. means) or other basic estimates (e.g. regression coefficient) AND variation (e.g. standard deviation) or associated estimates of uncertainty (e.g. confidence intervals) |
| ☐ | ☒ | For null hypothesis testing, the test statistic (e.g. *F*, *t*, *r*) with confidence intervals, effect sizes, degrees of freedom and *P* value noted<br>*Give P values as exact values whenever suitable.* |
| ☒ | ☐ | For Bayesian analysis, information on the choice of priors and Markov chain Monte Carlo settings |
| ☒ | ☐ | For hierarchical and complex designs, identification of the appropriate level for tests and full reporting of outcomes |
| ☒ | ☐ | Estimates of effect sizes (e.g. Cohen's *d*, Pearson's *r*), indicating how they were calculated |

*Our web collection on statistics for biologists contains articles on many of the points above.*

## Software and code

Policy information about availability of computer code

| | |
|---|---|
| Data collection | Juicebox (juicer_tools_0.7.5.jar)  was used for extracting hic chromosome matrices  from .hic files with conversion to .npz sparse matrices. Cooler package in python (cooler 0.8.10) was used for extraction of hic chromosome matrices from .cool files with conversion to .npz sparse matrices |
| Data analysis | Domain calls were made with 3dnetmod from https://bitbucket.org/creminslab/cremins_lab_tadsubtad_calling_pipeline_11_6_2021/.  loops on hic were called from custom python code based on HICCUPs approach from Aiden and colleagues available https://bitbucket.org/creminslab/cremins_lab_loop_calling_pipeline_11_6_2021/.  bowtie version 0.12.7 and samtools version 1.2 were used for Chipseq and Cut&Run fastq processing. MACS2 2.1.1.20160309 was used for peak calls.  bedtools 2.15.0 was used for intersection of peaks with domain boundaries.  opencv-python 4.2.0.32 (cv2 4.2.0) was used for APA pileup of domains. linalg class of numpy  (numpy 1.16.6) was used for eigenvector decomposition  and compartment calling in hic matrices. Custom scripts in python were used to access domain layer hierarchy and domain intersection with compartment and dots and subsequent boundary classification as well as IZ to closest boundary permutation test. For repliseq, the BIRCH clustering algorithm was used for IZ determination. |

For manuscripts utilizing custom algorithms or software that are central to the research but not yet described in published literature, software must be made available to editors and reviewers. We strongly encourage code deposition in a community repository (e.g. GitHub). See the Nature Research guidelines for submitting code & software for further information.

# Data

Policy information about availability of data

 All manuscripts must include a data availability statement. This statement should provide the following information, where applicable:
- Accession codes, unique identifiers, or web links for publicly available datasets
- A list of figures that have associated raw data
- A description of any restrictions on data availability

A full Data Access inventory is listed on p. 12 of the manuscript:

Data Availability Statement

>>All new raw data created in this manuscript has been uploaded to the 4D Nucleome portal and will be freely released for full distribution to the scientific community.

(1) Group 1: Sixteen-fraction Repli-seq data for H1 human ES cells:
- H1 human ES raw fastq: https://data.4dnucleome.org/experiment-sets/4DNESXRBILXJ/
- H1 human ES read depth normalized array for visualization: https://data.4dnucleome.org/files-processed/4DNFIEEYFQ7C/
- H1 human ES read depth scaled normalized array for IZ calls: https://data.4dnucleome.org/files-processed/4DNFI3N8GHKR/
- H1 human ES Early, Early-mid, Late IZs on read depth normalized array: https://data.4dnucleome.org/files-processed/4DNFIRF7WZ3H/

(2) Group 2: Sixteen-fraction Repli-seq data for Wild Type HCT116 cells:
- WT HCT116 raw fastq: https://data.4dnucleome.org/experiment-sets/4DNESNGZM5FG/
- WT HCT116 mitochondria normalized array for IZ calls: https://data.4dnucleome.org/files-processed/4DNFIPIQTMJ9/
- WT HCT116 Early, Early-mid, Late IZs on mitochondria normalized array: https://data.4dnucleome.org/files-processed/4DNFI95K53YS/

(3) Group 3: Sixteen-fraction Repli-seq data for wild type and cohesin knock-down HCT116 pairing:
- Rad21 knock-down HCT116 raw: https://data.4dnucleome.org/experiment-sets/4DNES92AU9JR/
- Rad21 knock-down HCT116 read depth normalized down sampled array for IZ calls: https://data.4dnucleome.org/files-processed/4DNFI3ZMWG5T/
- Rad21 knock-down HCT116 Early, Early-mid, Late IZs called on the read depth normalized down sampled array: https://data.4dnucleome.org/files-processed/4DNFIGOMS9G7/
- WT HCT116 raw fastq: https://data.4dnucleome.org/experiment-sets/4DNESNGZM5FG/
- WT HCT116 read depth normalized down sampled array for IZ calls: https://data.4dnucleome.org/files-processed/4DNFI6NGWNOG/
- WT HCT116 Early, Early-mid, Late IZs called on the read depth normalized down sampled array: https://data.4dnucleome.org/files-processed/4DNFIYO3H24N/

(4) Group 4: Sixteen-fraction Repli-seq data for wild type and WAPL knock-down HCT116 pairing:
- WAPL knock-down HCT116 raw: https://data.4dnucleome.org/experiment-sets/4DNES72NE7SL/
- WAPL knock-down HCT116 read depth normalized down sampled array for IZ calls: https://data.4dnucleome.org/files-processed/4DNFI7MI88QR/
- WAPL knock-down HCT116 Early, Early-mid, Late IZs called on the read depth normalized down sampled array: https://data.4dnucleome.org/files-processed/4DNFIDI1QJVA/
- WT HCT116 raw fastq: https://data.4dnucleome.org/experiment-sets/4DNESNGZM5FG/
- WT HCT116 read depth normalized down sampled array for IZ calls: https://data.4dnucleome.org/files-processed/4DNFI6NGWNOG/
- WT HCT116 Early, Early-mid, Late IZs called on the read depth normalized down sampled array: https://data.4dnucleome.org/files-processed/4DNFILNNSFMD/

(5) Group 5: Sixteen-fraction Repli-seq data visualization
- WT HCT116 read depth normalized down sampled array for visualization: https://data.4dnucleome.org/files-processed/4DNFI6NGWNOG/
- Rad21 knock-down HCT116 read depth normalized down sampled array for visualization: https://data.4dnucleome.org/files-processed/4DNFI3ZMWG5T/
- WAPL knock-down HCT116 read depth normalized down sampled array for visualization: https://data.4dnucleome.org/files-processed/4DNFI7MI88QR/

(6) Hi-C for wild type and WAPL knock-down HCT116 pairing:
- WAPL KD HCT116 raw Hi-C: https://data.4dnucleome.org/experiment-set-replicates/4DNES1JP4KZ1/
- WAPL KD HCT116 normalized balanced Hi-C matrices: https://data.4dnucleome.org/files-processed/4DNFIY5939F3/
- WAPL KD HCT116 Loops: https://data.4dnucleome.org/files-processed/4DNFILP7BD5H/
- WT HCT116 raw Hi-C: https://data.4dnucleome.org/experiment-set-replicates/4DNESNSTBMBY/
- WT HCT116 normalized balanced Hi-C matrices: https://data.4dnucleome.org/files-processed/4DNFI5MR78O6/
- WT HCT116 Loops: https://data.4dnucleome.org/files-processed/4DNFIOQLL854/

(7) Two-fraction Repli-seq data for human iPS wild type and two CRISPR engineered lines: raw data and processed log2(Early/Late) from three conditions
- WT human iPS line Raw Data: https://data.4dnucleome.org/experiment-sets/4DNESDYES9QD/
- WT human iPS line log2(Early/Late): https://data.4dnucleome.org/files-processed/4DNFI5WEY784/
- human engineered clone 1 80kb IZ deletion iPS line Raw Data: https://data.4dnucleome.org/experiment-sets/4DNESE3WCUAQ/
- human engineered clone 1 80kb IZ deletion iPS line log2(Early/Late): https://data.4dnucleome.org/files-processed/4DNFIZMB415V/
- human engineered clone 2 30kb control deletion iPS line Raw Data: https://data.4dnucleome.org/experiment-sets/4DNES66YWJU7/
- human engineered clone 2 30kb control deletion iPS line log2(Early/Late): https://data.4dnucleome.org/files-processed/4DNFIWDMF7HW/

(8) 5C data for human IPS wild type and two engineered lines:
primer bed file, raw heatmaps and processed heatmaps from three conditions
- WT human iPS line Raw Data: https://data.4dnucleome.org/experiment-set-replicates/4DNESLRDUPZ6/
- WT human iPS line balanced 5C data: replicate 1: https://data.4dnucleome.org/files-processed/4DNFIXM8V3ZB/; replicate 2: https://data.4dnucleome.org/files-processed/4DNFIDB6M1ZN/
- WT human engineered clone 1 80kb boundary deletion iPS line Raw Data: https://data.4dnucleome.org/experiment-set-replicates/4DNES39F1QWU/
- WT human engineered clone 1 80kb boundary deletion iPS line balanced 5C data: https://data.4dnucleome.org/files-processed/4DNFIA8P94BX/
- WT human engineered clone 2 30kb control deletion iPS line Raw Data: https://data.4dnucleome.org/experiment-set-replicates/4DNES3PDMUHG/
- WT human engineered clone 2 30kb control deletion iPS line balanced 5C data: replicate 1: https://data.4dnucleome.org/files-processed/4DNFI7WZYRHP/;

replicate 2: https://data.4dnucleome.org/files-processed/4DNFI7V4VXAQ/

>> 4D Nucleome data analyzed for this manuscript:
(1) Hi-C 2.5 data in H1 human ES cells:
https://data.4dnucleome.org/files-processed/4DNFI82R42AD/

(2) 16 fraction Repliseq on H1-hESC Tier1 cells:
https://data.4dnucleome.org/experiment-sets/4DNESXRBILXJ/

(3) Hi-C in untreated HCT116 Rad21-mAID cells:
https://data.4dnucleome.org/files-processed/4DNFIFLDVASC/

(4) Hi-C in auxin-treated for 360 minutes HCT116 Rad21-mAID cells:
https://data.4dnucleome.org/files-processed/4DNFILP99QJS/

(5) CTCF H1 human ES Cut&Run:
https://data.4dnucleome.org/experiment-set-replicates/4DNES1RQBHPK/

(6) Two-fraction Repli-seq for H1 human ES cells:
https://data.4dnucleome.org/files-processed/4DNFIISI1ZA8/

>>Processed data files for all Figures are provided as Supplementary Tables.

>>Reanalyzed data:
(1) CTCF H1 human ES Cut&Run:
https://data.4dnucleome.org/experiment-set-replicates/4DNES1RQBHPK/

(2) Rad21 human H9 ES ChIP-seq:
https://www.ncbi.nlm.nih.gov/geo/query/acc.cgi?acc=GSE105028/

(3) H1 human ES RNA-seq
https://www.encodeproject.org/experiments/ENCSR537BCG/

(4) Rad21 HCT116 ChIP-seq:
https://www.encodeproject.org/experiments/ENCSR000BSB/

(5) CTCF HCT116 ChIP-Seq
https://www.encodeproject.org/experiments/ENCSR000BSE/

(6) SNS-seq data:
https://www.ncbi.nlm.nih.gov/geo/query/acc.cgi?acc=GSE37757

(7) Hi-C WT HAP1:
https://ftp.ncbi.nlm.nih.gov/geo/series/GSE137nnn/GSE137372/suppl/GSE137372_hap1_wt_hic_20000_iced.matrix.gz
https://ftp.ncbi.nlm.nih.gov/geo/series/GSE137nnn/GSE137372/suppl/GSE137372_hap1_wt_hic_20000_ord.bed.gz

(8) Hi-C HAP1 CLONE 21:
https://ftp.ncbi.nlm.nih.gov/geo/series/GSE137nnn/GSE137372/suppl/GSE137372_hap1_clone21_hic_20000_iced.matrix.gz
https://ftp.ncbi.nlm.nih.gov/geo/series/GSE137nnn/GSE137372/suppl/GSE137372_hap1_wt_hic_20000_ord.bed.gz

(9) Hi-C in untreated HCT116 Rad21-mAID cells:
https://data.4dnucleome.org/files-processed/4DNFIFLDVASC/

(10) Hi-C in H1 human ES 2.5
https://data.4dnucleome.org/files-processed/4DNFI82R42AD/

(11) 16 fraction Repliseq on H1-hESC Tier1 cells
https://data.4dnucleome.org/experiment-sets/4DNESXRBILXJ/

>>Data links
(1) ORM Data
NCBI BioProject database accession PRJNA788726 http://genome.ucsc.edu/s/dsaulebe/ORM%20data%20HCT116

(2) Two-fraction Repli-seq data for Blobel engineered lines: raw data and processed log2(Early/Late) from three conditions
https://www.ncbi.nlm.nih.gov/geo/query/acc.cgi?acc=GSE190117

# Field-specific reporting

Please select the one below that is the best fit for your research. If you are not sure, read the appropriate sections before making your selection.

☒ Life sciences      ☐ Behavioural & social sciences      ☐ Ecological, evolutionary & environmental sciences

For a reference copy of the document with all sections, see nature.com/documents/nr-reporting-summary-flat.pdf

# Life sciences study design

All studies must disclose on these points even when the disclosure is negative.

| | |
|---|---|
| Sample size | N/A |
| Data exclusions | N/A |
| Replication | The early IZ enrichment findings were reproduced with both 16 fraction Repli-seq and SNS-seq. The early IZ peturbation result upon cohesin knock-down was reproduced with two-fraction Repli-seq, sixteen-fraction Repli-seq, and single-fraction early S phase BrdU labeling and pull-down. |
| Randomization | N/A |
| Blinding | N/A |

# Reporting for specific materials, systems and methods

We require information from authors about some types of materials, experimental systems and methods used in many studies. Here, indicate whether each material, system or method listed is relevant to your study. If you are not sure if a list item applies to your research, read the appropriate section before selecting a response.

### Materials & experimental systems

| n/a | Involved in the study |
|---|---|
| ☒ | ☐ Antibodies |
| ☐ | ☒ Eukaryotic cell lines |
| ☒ | ☐ Palaeontology and archaeology |
| ☒ | ☐ Animals and other organisms |
| ☒ | ☐ Human research participants |
| ☒ | ☐ Clinical data |
| ☒ | ☐ Dual use research of concern |

### Methods

| n/a | Involved in the study |
|---|---|
| ☒ | ☐ ChIP-seq |
| ☒ | ☐ Flow cytometry |
| ☒ | ☐ MRI-based neuroimaging |

## Eukaryotic cell lines

Policy information about cell lines

| | |
|---|---|
| Cell line source(s) | H1 human ES cells, HCT116 cell line |
| Authentication | All cell lines and their stocks and their culture protocols have been authenticated by the 4D Nucleome and 4D Nucleome standards were used. |
| Mycoplasma contamination | The lines have been verified via the 4D Nucleome. |
| Commonly misidentified lines (See ICLAC register) | N/A |

