## [Peer Review File · Nature]

Manuscript Title: Cohesin-mediated loop anchors confine the location of human replication origins

Redactions – Third Party Material: Parts of this Peer Review File have been redacted as indicated to remove third-party material.

Reviewer Comments & Author Rebuttals

Reviewer Reports on the Initial Version:

Referee #1 (Remarks to the Author):

Emerson et al. approach a very important question in the field of DNA replication, namely what are the determinants of the position of the initiation zones (IZ).

The authors apply a new computational tool developed by the group to classify the nested structures identifiable in high resolution HiC data. By this approach, they come to categorising 1. Non-compartment 2. Compartment domains. They focus on the former, whose boundaries are comprised within one compartment, and further distinguish domains with or without corner dot structures, where the corner dot is interpreted as signal generated by cohesin accumulated at strong boundary elements. Within each of the two categories, they further distinguished 3 types of domains depending on the level of nesting.

Next, the authors study the localisation of initiation zones with respects to the different type of domain boundaries identified above. The conclusion is that early initiation zones (IZ) are enriched in proximity of the boundaries of the corner-dot domains, while the late IZs are more frequently located in proximity of the boundaries of the dot-less domains. To investigate if cohesin play a role in this localisation, the authors employ a degron system, to trigger cohesin rapid degradation. Their depletion induces dispersion of the early initiation zones, indicating that cohesin participate to focusing the IZs distribution. Next, the authors explore the effect of cohesion depletion on the timing of activation of the IZs. After cohesin depletion, the signal of the early IZs becomes blurred over the 16 S-phase fractions collected, leading to the conclusion that IZ activation is delayed and that the activation of late IZs is even further delayed. In summary, according to the authors' interpretation, cohesin determine both the location and the timing of activation of the IZs.

Major points:

1. The two major points of this work are that cohesin control a. the distribution b. the timing of activation of IZs. The relationship between cohesin and origin of replication has already been explored in the paper Guillou et al., G&D 24, 2010. In this paper, the authors had already identified the enrichment of cohesin in proximity of origins and had found that depletion of cohesin had an effect on the efficiency of origin firing. Although Ermerson et al. add to these initial studies, availing of new technology and analysis genome-wide, this original work should have been cited and recognised.

2. The identification of IZs by high resolution repli-seq relies on the enrichment of reads in the

proximity of initiation sites as compared with further away, within approximately one replication domain (16 fractions). The authors of this study first suggest that the deletion of cohesin, and the consequent loss of the cohesin-dependent boundaries, causes a delocalisation of the IZs, leading to a dispersion over longer chromatin range (hence the blurred signal in Fig. 3g). But if we accept this interpretation of the data (which I find plausible), there is a fundamental problem in analysing the timing of activation of IZs across the different fractions. As the IZ signal is spatially dispersed, it is impossible to then draw any conclusion about the timing of activation, unless a different method is used to validate the results.

3. Figure. 3k is not cited in the main text and it is, instead, much more informative than the aggregated plots. In the cohesin depleted sample, the signal corresponding to the IZ in the wild type is split over may be 3 or 4 fractions, one being the same as the wild type, the others later. However, especially on the left, the signals flanking the apex seem mostly concentrated in the same fraction as in the wild type. Shouldn't a dispersion of the initiation events across the fractions be reflected in a shift of the neighbouring regions? I would expect the shape of the curve to change or at least become much more blurred. This image is difficult to reconcile with the strong impact that the authors attribute to cohesin depletion, and it is instead more compatible with the milder effect that is seen at a population level. If cohesin depletion decreases the efficiency of firing (as suggested by Guillou et al.), due to loss of anchoring to the boundary, then it is possible to envisage a scenario where some of the strongest origins could fire, although within a wider time-window, as the data presented suggest, while the weaker would possibly not fire at all, leading to an increase in inter-fork distances (Guillou et al.). Ultimately, however, the effect would be minor in terms of timing of replication within a whole population, leading thus to an unchanged overall replication timing. If this interpretation is correct, the effect is interesting, but for a more specialised audience.

4. Supplemental Fig. 12 c and d are really difficult to interpret. In c there seem to be no appreciable differences. Accordingly, in d, the range of the values is so small ($\log_2 = \pm 0.2$, equivalent to a 1.15 ratio!) that I am not sure what is the significance. I cannot conclude from this figure that there is any difference deleting cohesin in G1 or S.

5. In Fig. 4, it is not clear if the partially deleted IZ remains active or not. How can the authors be sure that the delayed replication is not a consequence of losing the activity of the origin, the region now being passively replicated by a fork emanating from a later origin?

In conclusion, while I am convinced that cohesin play a role in directing the localisation of the IZs with respect to the domain boundaries, I am not convinced about the effect on the timing of replication. My doubts are also reflected by the fact that, ultimately, the replication-timing profile does not show major changes. How do the authors explain this? If cohesin were to have a global role in the regulation of timing of IZ activation, their depletion would certainly have a much stronger impact on the replication-timing profile, even at a low resolution. I would expect that the whole profile of distribution of the signal of the BrdU-IPed DNA over the fractions should be altered more profoundly.

Based on my comments above I do not recommend the manuscript to be accepted for publication in Nature. I think the data are interesting and the question important. After strengthening the conclusion on the timing by employing a different method and writing the manuscript more clearly, it should be submitted to a more specialised journal.

Minor points

1. The paper is really difficult to read. Better written text and figure legends could help immensely. Too much space is dedicated to the classification of the different domains. Conceptually, this part is not particularly novel and eats into the second part of the manuscript that is potentially more impactful and requires much better explanations.
2. Have the authors checked the efficiency of BrdU incorporation upon cohesin depletion? Guillou et al. reported that Rad21 siRNA reduced BrdU incorporation, and this could skew the results of the repli-seq. The mitochondrial DNA, used as normalisation factor for the IP, is likely unaffected. This could lead to the decreased amplitude of the signal of the repli-seq, both at low and high resolution.
3. In the synchronisation experiments there is no FACS data shown. This would be important.
4. What are Figure 3c and f showing? There are no coordinates and the images above and below are aggregated plots.

Referee #2 (Remarks to the Author):

In the paper by Emerson et al, entitled "Cohesin-mediated loop anchors confine the localisation of human replication origins", the authors address a possible mechanism involved in the previously identified correlation between genome 3D organization and replication time. To do so, the authors combine standard bioinformatics analysis with degron and CRISPR experiments to dissect the role of G1 TADs and loops in the activation of initiation zones for genome replication. With the analysis and experiments, the authors conclude (and to my view demonstrate) that cohesin-mediated loops encoded in TADs (or subTADs or "loopy domains") are markers for localizing replication origins in S phase. Overall, I found the manuscript easy to read and the described work well conducted. The executed experiments were properly designed to address the questions at hand. Therefore, I have no major criticisms in this regard. However, I found that some of the findings were not discussed in light of previously published works, which would balance their novelty. Some specific comments are outlined below:

- The hierarchical (or nested) organization of chromatin domains have been already explored in many instances. Moreover, the classification of loopy versus non-loopy TADs was already proposed years ago with the first kilobase resolution Hi-C maps. Therefore, to this reader the classification of domains into 12 groups needs to be placed in the context of previous classifications.
- Along these lines, in several articles including seminal ones by co-authors of this work, related compartmentalization with early/late replication in the genome. Here, the results show again this correlation by makes emphasis on TAD/subTAD domains in contrast to compartment domains. This would require some explanation/discussion so the message to the reader is clearer.
- One strong message is that loopy TADs are associated to early replication. The authors made an effort to show that the definition of loops by their method, and the parameters used in the loop caller, have limited effect on the TAD classification and final conclusions of the work. However, it is known that loop calling is an open problem in the field and that many different methods will give different results. Have the authors used alternative approaches to call for loops (beyond their own

or the close one HICCUPS)? What would be the effect?

- I would have expected a more detailed analysis of the large differences in numbers for each of the 12 domains types in the different cell types. These are very different and few comments on the result would have been welcomed.
- The CRISPR editing of the 80 kb region with 10 CTCF sites removes part of the Early Wave IZ. The control CRISPR removes 30 kb downstream loop no near by the IZ. I miss more discussion of the so different perturbation of the CRISPR and the possible effects. Moreover, the CRIPR region is very close to a region where the Hi-C maps show unmappability (at least in the figures), which makes the interpretation more complex. Could this be further discussed?
- The finding that large majority of early IZ contain multiple CTCF sites is somehow circular as they are enriched in boundaries of loopy TAD/subTAD, which by definition are in CTCF sites forming loops. This is somehow expected and may need to be pointed out in the text.
- Finally, and possibly beyond the scope, I wonder whether it could be possible to track under the microscope the diffusion of cohesin over two detected TAD/subTAD weak and strong boundaries. This would validate the proposed model.

Other minor points.

- Figure 1. I would make all Hi-C matrices (in fact in all figures) perfectly squared. Use white font numbers in vertical in panel j. Change “Llayer” for “Layer” in the legend.
- Figure 2. Panel d. Why late IZs distributions for groups 1-3 are not mark as significant? They look to me very significant.

Referee #3 (Remarks to the Author):

In their manuscript 'Cohesin-mediated loop anchors confine the location of human replication origins' Emerson and coworkers reveal the general position of replication initiation zones. Early initiated zones appear to be at sites of divergent CTCF sites. The authors come to this conclusion by first making a classification of the genome based on Hi-C data, and then they make a second classification for TAD boundaries. Early S phase origins turn out to lie at TAD boundaries with a corner dot. These generally are strong boundaries, bound by CTCF and cohesin. Depletion of cohesin results in a delay in replication timing, indicating that correct 3D genome organization is required for replication initiation. Using a more local approach, the authors show that removal of divergent CTCF consensus sites alters replication timing of the respective locus.

My opinion about this paper is two-sided. On one hand, the authors perform 16 fractions of RepliSeq, which is nice but not quite new. I am also no fan of the way the paper has been written, as the classification of the genomic analyses takes up a massive portion of the available text, of which I wonder whether this is really required.

On the other hand, the finding that early S phase starts firing origins at divergent CTCF sites is fascinating, especially with the observation that the local removal of CTCF sites leads to a change in replication timing. The dependency of replication timing on cohesin is also really interesting.

Major points:

- 1) To my opinion, this manuscript could do with a major rewrite. The first couple of pages of the

manuscript are used to explain the classification of TADs, are on how this is only relevant for Hi-C and not micro-C data, and on which thresholding needs to be used. After 7 pages, the authors then make a new classification based on the boundaries of TADs, which then ultimately is used to come to conclusions. This paper presents some pretty interesting findings, but these are really difficult to distill from this lengthy whole. I would recommend that the authors strongly trim down on their genomic classification section. This will be needed to maintain the attention of the readers until the point that the authors start drawing their conclusions.

2) Related to point 1: surely figure 1 should include suppl. Table 3, as this gives a clear and digestible overview of all the classes? Likewise, the boundary classification could also do with a figure to make it understandable.

3) Several divergent CTCF sites together often result in a strong boundary. It would therefore be good to look into the correlation between boundary strength and replication timing.

4) Figure 2: I don't see a difference between class 3 and 6 regarding the presence of a corner-dot. In both classes this is present with a similar strength, and I don't understand the explanation in the text. If none of the individual sites show an enrichment at the corner, then how can an aggregate analysis show a corner-dot? Did the authors run an APA on these classes and see a difference in intensity? Otherwise it is a bit odd to call it dot-less. Or does the differences between the classes lie in the boundary strength? Then surely this should be the way to classify?

5) Figure 5: The blue rectangles indicating initiation zones run through the ChIPseq tracks, but do not at both sites touch CTCF/cohesin peaks. This does not correspond with the conclusions. Please explain.

Minor points:

- The annotation within the Hi-C maps is so large that the data in some cases has become invisible. Take for example the annotation of the corner-dots in figure 5a, which de facto mask everything. Or e.g. in figure 2a, the shape of the TAD of the "second 1st wave" is invisible in the mirror-image due to all the annotations.

- The use of the phrase 'no compartment' is confusing. A TAD is either in the A or B compartment.

Author Rebuttals to Initial Comments:

Overview for Reviewers

We want to thank all three reviewers for their overall positive feedback on the impact of the work, as well as their critical insights into ways to improve the writing, the analysis, and the clarity of insights gleaned from the data. Below, we carefully address each critique. Our resubmitted manuscript is significantly improved as a result of the helpful comments. As recommended by 2/3 Reviewers, we did decide to overhaul the writing of the story. In the process of addressing Reviewer feedback we discovered a TAD boundary stratification that directly led to a clean result based on the motifs at the boundaries. In addition, we add a large amount of new data. Specifically, we now add a newly published technique, optical replication mapping (ORM), for independently mapping replication initiation in a single-molecule manner. Using this technique completely orthogonal to 16-fraction Repli-seq, we exactly recapitulate the strong enrichment of replication initiation zones (IZs) at a key subset of genetically-encoded complex CTCF+cohesin boundaries.

In the 3D genome field, technological advances over a decade have resulted in an explosion of progress in terms of creating genome-wide ultra-high-kilobase-resolution maps of chromatin folding. We now have in hand the genome-wide locations of a full cohort of A/B compartments, TADs, subTADs, and loops across many mammalian cell types. However, all of these folding patterns are descriptive and there is a great need for perturbative experiments and a deep understanding of how 3D genome patterns influence genome function.

Here we provide an exciting cohort of new functional data, new analyses, and reformatted figures in which we take the following approach:

Below I outline the scope of the changes we have made in the resubmission:

1. We overhaul the writing of the story to center the genome's structure-function relationship
2. In response to Reviewer critique, we have rewritten the paper and reformatted and streamlined the analyses into 4 new main Figures and 14 new Supplementary Figures
3. We identify TADs, subTADs, and loops genome-wide using a novel Hi-C dataset from H1 human ES cells created under experimental conditions amenable to detecting 2-4-fold more loops than conventional Hi-C in pluripotent stem cells. This so-called Hi-C_2.5 data was generated by the Dekker lab in the 4DN Consortium and ultimately provided a unique window into loops genome-wide due to the special set of crosslinking conditions amendable to a 2-4-fold increase in sensitivity for capturing and detecting loops in the pluripotent human ES cell fate. Because we redid all reanalyses with this higher resolution dataset strengthened our conclusions.
4. We formulated new structural boundary classes based on loops, CTCF, cohesin, and CTCF motif orientation. This led to the clean identification of one specific genetically-encoded boundary type that informs the genomic localization of replication origins, addressing reviewers' confusion regarding the large number of boundary classes described in the first submission.
5. To prove that our results were reproducible, we added a newly published technique, optical replication mapping (ORM), for independently mapping

replication initiation in a single-molecule manner. Using this technique that is orthogonal to 16-fraction Repli-seq, we recapitulate the strong enrichment of replication initiation zones at a key subset of genetically-encoded complex CTCF+cohesin boundaries.

6. We provide an exciting cohort of new functional data:

→ **Global loss-of-structure experiments:** global knock-down of cohesin to assess if loops at complex CTCF motif boundaries are necessary for the location of replication initiation genome-wide

- We have improved the 16-fraction Repli-seq analysis
- We also show with the orthogonal single-molecule technique that early IZs indeed undergo a striking diffusion in their localization at boundaries upon knock-down of Rad21

→ **Global gain-of-structure experiments:** global knock-down of the cohesin unloading factor WAPL to assess if loop strengthening at complex CTCF motif boundaries is sufficient for the location of replication initiation genome-wide

- We added a new 16-fraction Repli-seq data cohort and analyses
- We also added a new Hi-C data cohort and analyses

→ **Local loss-of-structure experiments:** CRISPR experiments to directly engineer specific loops at a complex CTCF motif boundary and ascertain if specific loops are necessary for the specific localization of a high-efficiency IZ

→ **Local gain-of-structure experiments:** Sleeping beauty boundary engineering experiments to directly engineer two specific boundaries and ascertain if they are sufficient for the specific placement of a high-efficiency IZ

- We added a new 2-fraction Repli-seq data cohort and analyses
- We also added a new Hi-C analyses

Altogether, we have demonstrated with multiple orthogonal methods that high-efficiency IZs are located precisely at the boundaries between dotted TADs with a high-density of co-localized CTCF/cohesin sites with motifs pointed in complex orientations. Late IZs are localized to TAD boundaries devoid of CTCF. Through both local and global perturbations - both breaking and building loops - we see that cohesin extrusion into complex CTCF motif orientation boundaries is both (1) necessary for the precise localization of high-efficiency IZs genome-wide as well as the establishment of low-resolution Repli-seq patterns consistent with the destruction of an early IZs and (2) sufficient for the narrowing of the location of existing early IZs genome-wide as well as the establishment of low-resolution Repli-seq patterns consistent with new early IZs.

We sincerely thank the reviewers for their feedback. Our point-by-point responses are included below in blue font, with inline text from the new manuscript bolded.

Best,

Jennifer E. Phillips-Cremins,

David M. Gilbert

on behalf of all authors in the Chen, Cremins, Dekker, Gilbert, and Yue labs

Referees' comments:

Referee #1 (Remarks to the Author):

Emerson et al. approach a very important question in the field of DNA replication, namely what are the determinants of the position of the initiation zones (IZ). The authors apply a new computational tool developed by the group to classify the nested structures identifiable in high resolution HiC data. By this approach, they come to categorising 1. Non-compartment 2. Compartment domains. They focus on the former, whose boundaries are comprised within one compartment, and further distinguish domains with or without corner dot structures, where the corner dot is interpreted as signal generated by cohesin accumulated at strong boundary elements. Within each of the two categories, they further distinguished 3 types of domains depending on the level of nesting.

Next, the authors study the localisation of initiation zones with respects to the different type of domain boundaries identified above. The conclusion is that early initiation zones (IZ) are enriched in proximity of the boundaries of the corner-dot domains, while the late IZs are more frequently located in proximity of the boundaries of the dot-less domains. To investigate if cohesin play a role in this localisation, the authors employ a degron system, to trigger cohesin rapid degradation. Their depletion induces dispersion of the early initiation zones, indicating that cohesin participate to focusing the IZs distribution. Next, the authors explore the effect of cohesion depletion on the timing of activation of the IZs. After cohesin depletion, the signal of the early IZs becomes blurred over the 16 S-phase fractions collected, leading to the conclusion that IZ activation is delayed and that the activation of late IZs is even further delayed. In summary, according to the authors' interpretation, cohesin determine both the location and the timing of activation of the IZs.

Comment1.1: We thank the reviewer for this concise summary of our first submission. One critical point up front is that we mistakenly used poor wording and gave the impression that we were concluding that Corner-Dot boundaries influence the regulation of replication timing. We take full responsibility and apologize – and we have fully overhauled and re-written the manuscript for clarity, power, and precision. We clarify throughout the new version that our model, and the data we use to test the model, is consistent with the role for Dot boundaries harboring CTCF+cohesin sites in complex orientations in the localization of IZs. We remove all claims related to timing, other than to say that IZ timing is indicative of their efficiency (early IZs = IZs that fire at high-efficiency; late IZs = IZs that fire with poor efficiency). We also clarify that when we see the distribution of replication timing changes, it is an indirect effect of the distance of sequences from the nearest origin is becoming altered by the de-localization of IZs due to cohesin knock-down or further narrowing of IZ localization due to WAPL knock-down.

We make the following explicit statement in the text on p. 15 during our discussion of the model:

We note that we do not see evidence for a direct role for cohesin on replication timing. In our model, cohesin-mediated loop extrusion is required for IZ placement, and we anticipate that any change on replication timing would be

subtle and only indirect due to the altered distance of nearby genomic regions to the nearest initiation site.

We also address this issue throughout the manuscript with changes as described below.

Major points:

1. The two major points of this work are that cohesin control a. the distribution b. the timing of activation of IZs. The relationship between cohesin and origin of replication has already been explored in the paper Guillou et al., G&D 24, 2010. In this paper, the authors had already identified the enrichment of cohesin in proximity of origins and had found that depletion of cohesin had an effect on the efficiency of origin firing. Although Ermerson et al. add to these initial studies, availing of new technology and analysis genome-wide, this original work should have been cited and recognised.

Comment1.2: We definitely were remiss in not citing Guillou et. al. and we thank the reviewer for pointing this out. Guillou et al., 2010 was prescient in premise and performed important early work relating cohesin to origin regulation. Although this study was limited to only microarray data from 1% of the genome, the early data is used to show enrichment of cohesin at sites of origins. The authors also use colP between cohesin and MCM complexes to support their direct interaction. The authors subsequently measured inter-origin distance by DNA combing and indirectly by replication foci intensity by microscopy in cohesin KD cells. No 3D genome work was performed or queried at that time as the technological advances were not yet available.

We have not only referenced this paper all throughout our manuscript, but we also state explicitly on p. 8 that the initial co-localization of cohesin and origins was reported by this work:

A prior report, using ENCODE phase I pilot microarray data of 1% of the human genome, reported enrichment of Rad21 at approximately 300 replication origins²⁸.

We believe our work provides major and significant conceptual and technological advances building on the foundation of Guillou et al. First, a major advance of our work is that we directly queried the localization of IZs upon the removal of cohesin, as well as the overexpression of cohesin, using both global and local perturbative experiments (Figures 2,3,4). Second, due to major technological advances, we query finely mapped origin/IZ locations genome-wide across multiple cell types using both Repli-seq and Optical Replication Mapping (ORM) (Figure 2e). Third, we not only provide perturbation of cohesin loss, but we also conduct experiments for global cohesin gain (i.e. gain-of-looping WAPL knock-down experiments), as well as targeted gain-of-boundary engineering, which allows for the first time (to our knowledge) the assessment that cohesin-mediated loop extrusion is not only necessary but also sufficient for IZ placement on the genome (Figures 3,4). Fourth, we incorporate a full cohort of novel 3D genome folding data and features to identify specific boundaries that predict the subset of cohesin binding sites which ultimately

will influence replication initiation. Guillou et al was published at a time before Hi-C and other Chromosome-Conformation-Capture sequencing technologies were available, and no 3D genome folding feature could have been incorporated into their manuscript or models other than conceptually (Figures 1,2,3,4). Fifth, we discover that cohesin binding sites alone are not enriched at early IZs, but rather they are co-bound CTCF/cohesin sites – and it is the interplay between the two which deterministically influence IZ placement (Figure 1). Sixth, we integrate IZs genome-wide as assessed by Repli-seq and Optical Replication Mapping (instead of a small selection of origins) with TADs, subTADs, loops, cohesin, and CTCF data at the bleeding edge of data technological advances (Figures 1,2,3,4).

Overall – given these six major advances – all with technology not used nor available at the time of publication of Guillou et al., 2010, we have presented observations and a new model and mechanism for IZ localization on the genome due to cohesin-based extrusion and stalling at complex CTCF motif encoded boundaries. Such a model would have been impossible to gain insight into in 2010 due to lack of major technological advances across multiple fields. Thus our work studies IZs genome-wide through the use of the latest genome-wide 3D genome experimental and computational technologies and followups to test the functionality and mechanisms behind such observations using perturbative experiments that engineer or ablate loops mediated by CTCF boundaries, cohesin extrusion, and cohesin-unloading by WAPL.

Importantly, we also note that with the data available at that time, Guillou et al., 2010 proposed a model in which cohesin acts by bringing together clusters of replication origins over long distance, which we respectfully point out is not consistent with the new observations afforded by the Hi-C genome folding patterns and the high-resolution genome-wide Repli-seq and Optical Replication Mapping data. If cohesin acts by facilitating the formation of replication foci containing many replication origins, by virtue of Guillou et al.,2010's conclusion we would only expect to see weakened initiation zones with no spreading of initiation signal outside of annotated wildtype initiation zones upon cohesin knock-down. However, in our work we observed initiation zone spreading that are specific to loop domain boundaries with cohesin+CTCF in convergent or divergent configuration. This supports our model in which cohesin focuses initiation factors by loop extrusion. To further support the loop extrusion model, we performed WAPL knock-down, which causes aberrant cohesin unloading and more numerous loops genome wide. We found that initiation zones are further restricted and manifest as narrower peaks (Supplementary Figure 13).

2. The identification of IZs by high resolution repli-seq relies on the enrichment of reads in the proximity of initiation sites as compared with further away, within approximately one replication domain (16 fractions). The authors of this study first suggest that the deletion of cohesin, and the consequent loss of the cohesin-dependent boundaries, causes a delocalisation of the IZs, leading to a dispersion over longer chromatin range (hence the blurred signal in Fig. 3g). But if we accept this interpretation of the data (which I find plausible), there is a fundamental problem in analysing the timing of activation of IZs across the different fractions. As the IZ signal is spatially dispersed, it is impossible to then draw any conclusion about the timing of activation, unless a different method is used to validate the results.

Comment1.3: Reviewer1 rightly points out that we made a conceptual error in the writing in the first submission and inadvertently created confusion about our interpretation of the data. The reviewer's interpretation is the same as our interpretation, and as stated in Comment1.1 we now clearly ensure that our claims are related to the role of cohesin extrusion and build-up at a key subset of boundaries on high-efficiency IZ localization and not replication timing per se. We do observe changes in the timing of replication of specific genomic bins, but we believe based on the data that this does not invoke any change in the average firing time of the origins but rather the positions of the origins, which then indirectly effects timing of specific genomic bins based on their distance from the initiation events.

To clarify this point in the manuscript we have done the following:

(i) When introducing replication analyses following cohesin depletion, we have added the sentence:

“We note that we do not see evidence for a direct role for cohesin on replication timing. In our model, cohesin-mediated loop extrusion is required for IZ placement, and we anticipate that any change on replication timing would be subtle and only indirect due to the altered distance of nearby genomic regions to the nearest initiation site.”

(ii) We have performed Optical Replication Mapping to map the positions of origins at higher resolution and accurately assess their efficiency upon cohesin knock-down. The results of this experiment also show that origins are de-localized, resulting in a loss of efficiency of origins at the centers of IZs and an increase in efficiency surrounding the IZ. Thus, de-localization can account for our results without invoking an effect on replication timing.

3. Figure 3k is not cited in the main text and it is, instead, much more informative than the aggregated plots. In the cohesin depleted sample, the signal corresponding to the IZ in the wild type is split over may be 3 or 4 fractions, one being the same as the wild type, the others later. However, especially on the left, the signals flanking the apex seem mostly concentrated in the same fraction as in the wild type. Shouldn't a dispersion of the initiation events across the fractions be reflected in a shift of the neighbouring regions? I would expect the shape of the curve to change or at least become much more blurred. This image is difficult to reconcile with the strong impact that the authors attribute to cohesin depletion, and it is instead more compatible with the milder effect that is seen at a population level. If cohesin depletion decreases the efficiency of firing (as suggested by Guillou et al.), due to loss of anchoring to the boundary, then it is possible to envisage a scenario where some of the strongest origins could fire, although within a wider time-window, as the data presented suggest, while the weaker would possibly not fire at all, leading to an increase in inter-fork distances (Guillou et al.). Ultimately, however, the effect would be minor in terms of timing of replication within a whole population, leading thus to an unchanged overall replication timing. If this interpretation is correct, the effect is interesting, but for a more specialised audience.

Comment1.4: We believe that this comment derives from the same confusion we created over whether the effect is on replication timing or origin positions, which we

have clarified above. We are in agreement with the reviewer that the changes in initiation zone features upon cohesin knock-down are primarily x-axis wise diffusion and are consistent with loss of precision in origin positioning (Figure 2A-D). To obtain more direct measurements of origin de-localization, we have performed single-molecule Optical Replication Mapping and found the signal within initiation zones becomes attenuated while aberrant signal immediately outside of initiation zones arises upon cohesin knock-down only at one key class of boundary (Class 1: Dotted TAD/subTAD boundaries with convergent and divergent CTCF+cohesin co-bound motifs) (Figure 2E). We also demonstrate that reducing cohesin removal results in a tightening of IZs, providing supportive evidence that the effect is indeed due to cohesin-mediated extrusion (Figure 3). Overall we conclude that the primary consequence of cohesin-mediated extrusion replication is localized high-efficiency initiation zones at boundaries with extremely strong insulation in wild type cells due to specific complex convergent/divergent CTCF motif orientation.

Reviewer 1's feedback has helped immensely in improving the manuscript. There is one point on which we disagree – we believe our observation that cohesin is necessary and sufficient to position replication origins reaches the impact of the broadest possible audience. Determinants of origin positioning in mammalian cells remains an unsolved mystery in molecular biology, of interest to broad audiences, and has remained elusive. The biological functions of loop extrusion and TAD boundaries are also a major mystery of broad interest across the fields of genetics, epigenetics, genome biology, and systems biology. Here we provide unique observations in both local and global gain- and loss-of-structure perturbations uncovering that a unique genetically-encoded boundary feature deterministically informs the functional positioning of origins.

Finally, in response to Reviewer 1's suggestion to include more visualizations, we have added significantly more heatmap examples in addition to the aggregate plots, including two specific examples now prominently in Figures 3A+3B and two specific examples in Supplementary Figure 14.

4. Supplemental Fig. 12 c and d are really difficult to interpret. In c there seem to be no appreciable differences. Accordingly, in d, the range of the values is so small ($\log_2 = \pm 0.2$, equivalent to a 1.15 ratio!) that I am not sure what is the significance. I cannot conclude from this figure that there is any difference deleting cohesin in G1 or S.

Comment1.5: We agree with the Reviewer's point. Given that we have doubled the amount of data and analysis in the story, we elected to remove this supplemental figure given that all of the new gain-of-structure, boundary engineering, and ORM data was so clear.

5. In Fig. 4, it is not clear if the partially deleted IZ remains active or not. How can the authors be sure that the delayed replication is not a consequence of losing the activity of the origin, the region now being passively replicated by a fork emanating from a later origin?

Comment1.6: The direct overlap of IZs with boundaries is not amenable to the use of engineering approaches to fully decouple them, and this remains a technical

challenge for the genome biology field at large for nearly every structure-function scenario. For example, enhancers directly overlap loops, so the best scenario current technology allows is the editing of the loop while destroying as little of the enhancer as possible (and vice versa) – and clearly stating the caveats. A common strategy used is to pair such perturbative loss-of-looping data with a gain-of-looping experiment. Similarly, with replication origins or IZs – there is partial overlap with loop anchors and the CTCF sites forming the loop. The best technology will allow at this time is to choose an IZ that only partially overlaps a boundary. And then pair this loss-of-boundary approach with a gain-of-boundary technique, thus asking if the boundary is both necessary and/or sufficient for the localization of an IZ. We now present a fully revised Figure 4 where we have added the following experiments, analyses, and text edits to address the important Reviewer 1 feedback. We also add all of these points to the text in the last 2 paragraphs on p. 14.

Revised Figure 4a-b: Local loss-of-structure experiments: We clarified in the text and Figure 4b that we used targeted CRISPR/Cas9 genome editing to delete an 80 kb section of the genome containing a complex array of more than 10 CTCF sites with both upstream and downstream motif orientations. We now clarify in the text, and also make Figure 4a-b clearer that this loop anchor was chosen because it *only partially overlaps* an early IZ, thus allowing us to ablate the loop while keeping much of the genome placement of the IZ intact. Importantly, we observed a striking local shift of $\log_2(\text{early/late})$ signal from early to late S phase upon deletion of the 80 kb loop anchor, which is consistent with the loss of an early IZ. Importantly, we also deleted a different boundary that did not overlap an IZ and we saw no shift in $\log_2(\text{E/L})$ signal. This experiment is of course not perfect for the reasons stated here, and we provide this clear realistic statement in the text on p. 14:

“The direct overlap of IZs with boundaries is not amenable to the use of engineering approaches to fully decouple them, and this remains a technical challenge for the genome biology field at large. Nevertheless, our data provide evidence that replication at a specific early IZ can undergo a striking shift to late S phase upon ablation of a boundary. These data are consistent with our global cohesin knock-down observations and our model that complex CTCF orientation boundaries are necessary for the placement of high-efficiency IZs.”

New Figure 4c-d: Local gain-of-structure experiments: Given that looping perturbative experiments have specific challenges related to decoupling them from the underlying chromatin features, we paired this data with extensive new experiments and analyses in which we have evaluated the effects of inserting a boundary into several sites in the genome on $\log_2(\text{E/L})$ signal from two-fraction Repli-seq. We used published cell lines from one of our previous collaborative studies (see Di Zhang, Jennifer Phillips-Cremins, Gerd Blobel, Nature Genetics, 2020) in which the sleeping beauty system was used to engineer specific boundaries at key locations in the genome. Using two-fraction Repli-seq data and re-analyzing Hi-C data in these lines, we again observed that a known sequence containing CTCF binding sites is *sufficient* to create new de novo boundaries and ectopically shift $\log_2(\text{E/L})$ signal from late to early S phase. These data are consistent with our global WAPL knock-down observations and our model that complex CTCF orientation boundaries are sufficient for the specific genomic

placement of high-efficiency IZs. We also added the following important section of text:

“Because the direct overlap of IZs with boundaries is not amenable to clear, single-variable “loss-of-structure” perturbative experiments, we also examined a “gain-of-structure” approach in which we engineered an ectopic boundary and assessed if it was sufficient to induce changes in replication initiation. We mapped replication with two-fraction Repli-seq in published HAP1 cell lines in which we have previously demonstrated a gain in boundary upon insertion of an established 2 kb-sized cell type-invariant boundary element⁴³. We observed a striking shift from late to early replication directly at the location of the engineered boundary (Figure 4d), consistent with the possibility that boundaries can be sufficient for de novo early IZ firing. Together, our data reveal that both global and local gain- and loss- of structural boundary engineering can deterministically influence the placement of IZs.”

In conclusion, while I am convinced that cohesin play a role in directing the localisation of the IZs with respect to the domain boundaries, I am not convinced about the effect on the timing of replication. My doubts are also reflected by the fact that, ultimately, the replication-timing profile does not show major changes. How do the authors explain this?

Comment1.7: We have fully addressed this point in Comments1.1-1.6.

If cohesin were to have a global role in the regulation of timing of IZ activation, their depletion would certainly have a much stronger impact on the replication-timing profile, even at a low resolution. I would expect that the whole profile of distribution of the signal of the BrdU-IPed DNA over the fractions should be altered more profoundly.

Comment1.8: We do not claim that the global role of cohesin lies in the regulation of *timing* of IZ activation but rather the positioning of IZs. Our data and analyses clearly show significant enrichment of IZs specifically at dot domain boundaries with cohesin+CTCF in convergent and divergent configurations and the loss of the localization precision as a result of cohesin knock-down. Together, these data support our conclusion that the global role of cohesin with respect to IZ regulation is the specification of IZ positioning. Global replication timing profiles can be interpreted as a composite function of IZ positioning and IZ firing efficiency. By altering IZ positioning replication, timing is affected secondarily, which is consistent with the limited extent to which the timing profile is altered in the absence of cohesin. We also point out that we show that the effect of cohesin on origin localization is confined to a specific subset of TAD boundaries. Thus, the indirect effects on replication timing are not expected to be profound or genome-wide. Our new data, figure, and text changes to clarify and support this point are described in Comments1.1-1.6.

Based on my comments above I do not recommend the manuscript to be accepted for publication in Nature. I think the data are interesting and the question important. After strengthening the conclusion on the timing by employing a different method and writing the manuscript more clearly, it should be submitted to a more specialised journal.

Comment1.9: Please see Comments 1.1-1.8, as well as the opening summary on p.1-2 of the full scope of the work in our revision. We fundamentally disagree, as this is one of, if not the most, comprehensive papers functionally addressing the genome's structure-function relationship using both gain- and loss-of-function approaches to date. We answer a fundamental decades-old question of mechanisms encoding placement origin placement by way of the 3D genome and we find a clear functional role for a genome folding features, which is the leading topic in the chromatin field.

Minor points

1. The paper is really difficult to read. Better written text and figure legends could help immensely. Too much space is dedicated to the classification of the different domains. Conceptually, this part is not particularly novel and eats into the second part of the manuscript that is potentially more impactful and requires much better explanations.

Comment1.10: We have fully overhauled and revised the manuscript, as well as the figures, to tell a clear genome-structure-function replication origin story. Thank you for this feedback, as it fundamentally helped us create a manuscript we are incredibly proud to share with the scientific community.

2. Have the authors checked the efficiency of BrdU incorporation upon cohesin depletion? Guillou et al. reported that Rad21 siRNA reduced BrdU incorporation, and this could skew the results of the repli-seq. The mitochondrial DNA, used as normalisation factor for the IP, is likely unaffected. This could lead to the decreased amplitude of the signal of the repli-seq, both at low and high resolution.

Comment1.11: This is an important comment. We now cite and show several pieces of evidence that knock-down of RAD21 and WAPL does not significantly affect BrdU incorporation.

First, Oldach & Nieduszynski (<https://www.mdpi.com/2073-4425/10/3/196>) have published that, in the same cell line and synchronization scheme that we employ to knock-down RAD21, there is no effect on BrdU incorporation measured by FACS.

Second, we directly quantify BrdU-DNA immunoprecipitated in each fraction of S phase and we see minimal differences across the 16 fractions in WAPL and Rad21 knock-down HCT116 vs. WT HCT116. We show this data in Supplementary Figure 10.

Third, we note that Guillou et al used HeLa cells in their experiments and showed a slowing of S phase and diminished BrdU incorporation in Rad21 KD cells. Nevertheless, Nieduszynski's paper used HCT116 and showed no difference between control and cohesin KD in S phase progression. We now provide the equivalent results from our experiments, shown in Supplemental Figure 10, which confirm that in our hands there is also no effect of RAD21 KD on S phase progression in HCT116 cells. There could be a HeLa specific effect on BrdU incorporation efficiency.

Fourth, we demonstrate in our own hands that there is minimal effect on S phase progression by DNA content in both WAPL and Rad21 knock-down systems. We show this data in Supplementary Figure 10.

3. In the synchronisation experiments there is no FACS data shown. This would be important.

Comment1.12: We demonstrate in our own hands that there is minimal effect on S phase progression by DNA content in both WAPL and Rad21 knock-down systems. We show this data in Supplementary Figure 10.

4. What are Figure 3c and f showing? There are no coordinates and the images above and below are aggregated plots.

Comment1.13: We have overhauled all Figures for this revision to make them more intuitive and clearer. The equivalent plots for this question are now in Figure 2c&e where we show 16 fraction Repli-seq for WT HCT116 (Figure 2c) and Rad21 knock-down (Figure 2e). Each row represents a temporal fraction from S phase, there are 16 rows/fractions, and the Repli-seq signal plotted represents an average for that fraction (y-axis) in 50 kb bins across a +/-750 kb genomic distance centered on the midpoint of boundaries (x-axis). Sample sizes for each class are shown in Figure 2a. We have added this description to the caption to aide in clarity.

Referee #2 (Remarks to the Author):

In the paper by Emerson et al, entitled "Cohesin-mediated loop anchors confine the localisation of human replication origins", the authors address a possible mechanism involved in the previously identified correlation between genome 3D organization and replication time. To do so, the authors combine standard bioinformatics analysis with degron and CRISPR experiments to dissect the role of G1 TADs and loops in the activation of initiation zones for genome replication. With the analysis and experiments, the authors conclude (and to my view demonstrate) that cohesin-mediated loops encoded in TADs (or subTADs or "loopy domains") are markers for localizing replication origins in S phase. Overall, I found the manuscript easy to read and the described work well conducted. The executed experiments were properly designed to address the questions at hand. Therefore, I have no major criticisms in this regard. However, I found that some of the findings were not discussed in light of previously published works, which would balance their novelty. Some specific comments are outlined below:

- The hierarchical (or nested) organization of chromatin domains have been already explored in many instances. Moreover, the classification of loopy versus non-loopy TADs was already proposed years ago with the first kilobase resolution Hi-C maps. Therefore, to this reader the classification of domains into 12 groups needs to be placed in the context of previous classifications.

Comment2.1: We agree with the Reviewer that more clarity is needed regarding the boundary classes from our first submission. For the revision, we have elected to

streamline the story to focus on the genome's structure-function relationship with replication origins, as advised by Reviewers 1 & 3. We removed Figure 1 and the extensive boundary classifications as requested by the Reviewers and streamlined concepts across multiple figures to stratify clean boundary classes that focus the reader on the biological discoveries.

We now use only 2 structural boundary classes already established in the literature: adjacent Corner-Dot TADs/subTADs on one or both sides (Dot boundaries) or (2) adjacent Dotless TADs/subTADs on both sides (Dotless boundaries), and then we immediately focus in Figure 1 on further stratifying Dot and Dotless boundaries by those that colocalize with (i) two or more high-density of CTCF/cohesin binding sites pointed in complex divergent/convergent orientations, (ii) one or more CTCF/cohesin binding sites pointed in tandem orientation, (iii) 0 CTCF-bound motifs. See below excerpt illustrating the 6 boundary classes used in Figure 1/2/3. We fully reference the literature supporting the key papers underlying the choice of these 6 boundaries classes, and we focus the rest of the paper on understanding the replication IZ patterns and how they change across a range of local and global structural perturbations.

- Along these lines, in several articles including seminal ones by co-authors if this work, related compartmentalization with early/late replication in the genome. Here, the results show again this correlation by makes emphasis on TAD/subTAD domains in contrast to compartment domains. This would require some explanation/discussion so the message to the reader is clearer.

Comment2.2: We now include the following statement in the opening paragraph of the manuscript: “Waves of early and late replication correlate with A and B compartments, respectively, and the temporal transitions from early to late replication align with TAD boundaries^{1, 14-16}. However, the role for fine-scale genome folding (such as loops, subTADs, and TADs detectable in high-resolution Hi-C data) on IZ and origin locations upon entry into S phase has not been investigated.”

- One strong message is that loopy TADs are associated to early replication. The authors made an effort to show that the definition of loops by their method, and the parameters used in the loop caller, have limited effect on the TAD classification and final conclusions of the work. However, it is known that loop calling is an open problem in the field and that many different methods will give different results. Have the authors used alternative approaches to call for loops (beyond their own or the close one HICCUPS)? What would be the effect?

Comment2.3: We now have added an extensive analysis in which we explore a large search space of possible loop calling statistical methodologies in a new

Supplementary Figure 6. In the first revision we only adjusted parameters within the existing statistical framework. Here, we conducted a multi-month effort to explore many different statistical assumptions across a range of loop callers in a single variable manner.

In this work, we explored we explored the below, and detail the results in a full Supplementary Figure 6 as well as the Supplementary Methods:

- Option A (N=292475 loops genome-wide; N=280873 loops on all autosomes) shows the effect of only using a global 1D expected model.
- Option B (N=42661 loops genome-wide; N=41091 loops on all autosomes) switches the global 1D expected from Option A to use a local expected by applying a geometrical donut filter with $p=2/w=10$ parameters at distance scales > 200 kb and applies a geometrical upper-triangle filter (so-called triu filter) with $p=2/w=10$ parameters at distance scales ≤ 200 kb.
- In Option C (N=24046 loops genome-wide; N=23040 loops on all autosomes), geometrical vertical and horizontal filters are added to Option B to model both local TADs/subTADs and stripes in the expected background.
- Option D (N=16922 loops genome-wide; N=16248 loops on all autosomes) involves the removal of triu from Option C and thus a geometrical donut filter with $p=2/w=10$ parameters is applied at all distance scales.
- Option E (N=19071 loops genome-wide; N=18314 loops on all autosomes) adjusts the donut filter size in Option D by increasing p to 4.
- Options F&G: The donut filter size was further adjusted in Options F (N=23395 loops genome-wide; N=22437 loops on all autosomes) and G (N=25687 loops genome-wide; N=24620 loops on all autosomes) by increasing w in Option E to 14 and 16, respectively.
- Option H: The triU filter was reintroduced with with $p=4/w=16$ at distance scales ≤ 200 kb to Option G (with $p=4/w=16$ donut expected and vertical/horizontal expected filters at > 200 kb distance scales) to produce Option H (N=28665 loops genome-wide; N=27415 loops on all autosomes).
- Option I: For all options A-H, we used a Qvalue threshold of 0.05. In Options I (N=30085 loops genome-wide; N=28663 loops on all autosomes) and J (N=31508 loops genome-wide; N=29933 loops on all autosomes), we modified Option H by using qvalue thresholds of 0.1 and 0.3, respectively, to decrease dot calling stringency.

Ultimately, we discovered that changing the loop calls genome-wide can indeed alter the proportion of TADs/subTADs that become Dot vs. No Dot domains, however, it is clear that our biological finding of the enrichment of early IZs at Dot domains with complex CTCF orientations is fully robust to a wide range of statistical approaches and parameters used for loop calling.

• I would have expected a more detailed analysis of the large differences in numbers for each of the 12 domains types in the different cell types. These are very different and few comments on the result would have been welcomed.

Comment2.4: In our current manuscript we now note in the text how the boundary numbers change by cell type, and we comment on the differences in numbers among the 6 groups. Across every analysis we have done in every figure, the

observations remain the same – and the relationship between IZs and boundaries remains the same - even if the exact numbers of dots, TADs, and IZs might vary by cell type.

- The CRISPR editing of the 80 kb region with 10 CTCF sites removes part of the Early Wave IZ. The control CRISPR removes 30 kb downstream loop no near by the IZ. I miss more discussion of the so different perturbation of the CRISPR and the possible effects. Moreover, the CRIPR region is very close to a region where the Hi-C maps show unmappability (at least in the figures), which makes the interpretation more complex. Could this be further discussed?

Comment2.5: The reviewer is correct that it is difficult to interpret deletion analyses in our case since the IZs are partially localized to the very boundaries we are trying to delete. The same sentiment was shared by Reviewer 1, and we have detailed our discussion of the issue as well as the new experiments we have added. For the reviewer's convenience, we have copy-pasted the response from Comment 1.6 below. In brief, to tease apart the contribution of IZ sequences vs boundaries, we have evaluated the effects of inserting a strong dot domain, complex CTCF orientation boundary into several sites in the genome to demonstrate that the boundary insertion is sufficient to create a $\log_2(E/L)$ pattern consistent with the gain of an IZ at an ectopic location in the genome.

“””

Comment1.6: The direct overlap of IZs with boundaries is not amenable to the use of engineering approaches to fully decouple them, and this remains a technical challenge for the genome biology field at large for nearly every structure-function scenario. For example, enhancers directly overlap loops, so the best scenario current technology allows is the editing of the loop while destroying as little of the enhancer as possible (and vice versa) – and clearly stating the caveats. A common strategy used is to pair such perturbative loss-of-looping data with a gain-of-looping experiment. Similarly, with replication origins or IZs – there is partial overlap with loop anchors and the CTCF sites forming the loop. The best technology will allow at this time is to choose an IZ that only partially overlaps a boundary. And then pair this loss-of-boundary approach with a gain-of-boundary technique, thus asking if the boundary is both necessary and/or sufficient for the localization of an IZ. We now present a fully revised Figure 4 where we have added the following experiments, analyses, and text edits to address the important Reviewer 1 feedback. We also add all of these points to the text in the last 2 paragraphs on p. 14.

Revised Figure 4a-b: Local loss-of-structure experiments: We clarified in the text and Figure 4b that we used targeted CRISPR/Cas9 genome editing to delete an 80 kb section of the genome containing a complex array of more than 10 CTCF sites with both upstream and downstream motif orientations. We now clarify in the text, and also make Figure 4a-b clearer that this loop anchor was chosen because it *only partially overlaps* an early IZ, thus allowing us to ablate the loop while keeping much of the genome placement of the IZ intact. Importantly, we observed a striking local shift of $\log_2(\text{early/late})$ signal from early to late S phase upon deletion of the 80 kb loop anchor, which is consistent with the loss of an early IZ. Importantly, we also deleted a different boundary that did not overlap an IZ and we saw no shift in

log₂(E/L) signal. This experiment is of course not perfect for the reasons stated here, and we provide this clear realistic statement in the text on p. 14:

“The direct overlap of IZs with boundaries is not amenable to the use of engineering approaches to fully decouple them, and this remains a technical challenge for the genome biology field at large. Nevertheless, our data provide evidence that replication at a specific early IZ can undergo a striking shift to late S phase upon ablation of a boundary. These data are consistent with our global cohesin knock-down observations and our model that complex CTCF orientation boundaries are necessary for the placement of high-efficiency IZs.”

New Figure 4c-d: Local gain-of-structure experiments: Given that looping perturbative experiments have specific challenges related to decoupling them from the underlying chromatin features, we paired this data with extensive new experiments and analyses in which we have evaluated the effects of inserting a boundary into several sites in the genome on log₂(E/L) signal from two-fraction Repli-seq. We used published cell lines from one of our previous collaborative studies (see Di Zhang, Jennifer Phillips-Cremins, Gerd Blobel, Nature Genetics, 2020) in which the sleeping beauty system was used to engineer specific boundaries at key locations in the genome. Using two-fraction Repli-seq data and re-analyzing Hi-C data in these lines, we again observed that a known sequence containing CTCF binding sites is *sufficient* to create new de novo boundaries and ectopically shift log₂(E/L) signal from late to early S phase. These data are consistent with our global WAPL knock-down observations and our model that complex CTCF orientation boundaries are sufficient for the specific genomic placement of high-efficiency IZs. We also added the following important section of text:

“Because the direct overlap of IZs with boundaries is not amenable to clear, single-variable “loss-of-structure” perturbative experiments, we also examined a “gain-of-structure” approach in which we engineered an ectopic boundary and assessed if it was sufficient to induce changes in replication initiation. We mapped replication with two-fraction Repli-seq in published HAP1 cell lines in which we have previously demonstrated a gain in boundary upon insertion of an established 2 kb-sized cell type-invariant boundary element⁴³. We observed a striking shift from late to early replication directly at the location of the engineered boundary (Figure 4d), consistent with the possibility that boundaries can be sufficient for de novo early IZ firing. Together, our data reveal that both global and local gain- and loss- of structural boundary engineering can deterministically influence the placement of IZs.”

“””

- The finding that large majority of early IZ contain multiple CTCF sites is somehow circular as they are enriched in boundaries of loopy TAD/subTAD, which by definition are in CTCF sites forming loops. This is somehow expected and may need to be pointed out in the text.

Comment2.6: We thank the Reviewer for this important point. We agree that once we had the boundaries stratified by dots that the discovery of CTCF orientation would be the anticipated finding, and in the first submission this provide important

validation. In this revision, to build on the Reviewer's feedback, we have spent the last 4-5 months addressing this specifically by overhauling the boundary classes – merging Figure 2 and 4 ideas and concepts into a new class of 6 boundaries that parses by CTCF motif orientation and dot structure immediately in Figure 1. We are particularly excited because in this new set of analyses, Class 1 boundaries with dots on one or both sides and complex CTCF motif orientation are highly predictive for locations in the genome where IZs are destroyed upon cohesin knock-down and precisely refined in WAPL knock-down.

- Finally, and possibly beyond the scope, I wonder whether it could be possible to track under the microscope the diffusion of cohesin over two detected TAD/subTAD weak and strong boundaries. This would validate the proposed model.

Comment2.7: This is a very interesting idea. Although we agree that it is beyond the scope of the experiments described in this manuscript (scope detailed in p. 1-2), it would be exciting to see follow-up studies using live cell single molecule tracking of Mcm, Rad21, and WAPL, as well as other origin binding proteins.

Other minor points.

- Figure 1. I would make all Hi-C matrices (in fact in all figures) perfectly squared. Use white font numbers in vertical in panel j. Change “Llayer” for “Layer” in the legend.

Comment2.8: We have made this change.

- Figure 2. Panel d. Why late IZs distributions for groups 1-3 are not mark as significant? They look to me very significant.

Comment2.9: We now present completely new statistical tests at all the new boundaries, and the results are consistent with the first revision but streamline the story.

Referee #3 (Remarks to the Author):

In their manuscript 'Cohesin-mediated loop anchors confine the location of human replication origins' Emerson and coworkers reveal the general position of replication initiation zones. Early initiated zones appear to be at sites of divergent CTCF sites. The authors come to this conclusion by first making a classification of the genome based on Hi-C data, and then they make a second classification for TAD boundaries. Early S phase origins turn out to lie at TAD boundaries with a corner dot. These generally are strong boundaries, bound by CTCF and cohesin. Depletion of cohesin results in a delay in replication timing, indicating that correct 3D genome organization is required for replication initiation. Using a more local approach, the authors show that removal of divergent CTCF consensus sites alters replication timing of the respective locus.

My opinion about this paper is two-sided. On one hand, the authors perform 16 fractions of RepliSeq, which is nice but not quite new. I am also no fan of the way the paper has been written, as the classification of the genomic analyses takes up a

massive portion of the available text, of which I wonder whether this is really required.

Comment3.1: We have fully rewritten the manuscript as the reviewer has rightly recommended. The full overview of all the changes, new analyses and new experiments is detailed in full in p.1-2. We have added multiple new gain-of-structure perturbations, new Hi-C data, a cohort of all new analyses and figures. Moreover, to address the issue of Repli-seq raised above – we also add brand new Optical Replication Mapping – and we indeed see that it mirrors the findings in the paper.

On the other hand, the finding that early S phase starts firing origins at divergent CTCF sites is fascinating, especially with the observation that the local removal of CTCF sites leads to a change in replication timing. The dependency of replication timing on cohesin is also really interesting.

Comment3.2: Thank you for this feedback – now in the re-submission we have reinforced our conclusions regarding the de-localization of origins with gain-of-structure and loss-of-structure experiments and analyses to reinforce this new very exciting observation.

Major points:

1) To my opinion, this manuscript could do with a major rewrite. The first couple of pages of the manuscript are used to explain the classification of TADs, are on how this is only relevant for Hi-C and not micro-C data, and on which thresholding needs to be used. After 7 pages, the authors then make a new classification based on the boundaries of TADs, which then ultimately is used to come to conclusions. This paper presents some pretty interesting findings, but these are really difficult to distil from this lengthy whole. I would recommend that the authors strongly trim down on their genomic classification section. This will be needed to maintain the attention of the readers until the point that the authors start drawing their conclusions.

Comment3.3: We agree. We have fully rewritten the paper as detailed throughout the rebuttal.

2) Related to point 1: surely figure 1 should include suppl. Table 3, as this gives a clear and digestible overview of all the classes? Likewise, the boundary classification could also do with a figure to make it understandable.

Comment3.4: We agree. We have now added clear schematic in the first 3 figures showing the boundary classes. We have also added extensive written clarity throughout the manuscript, as well as a large cohort of tables for the scientific community.

3) Several divergent CTCF sites together often result in a strong boundary. It would therefore be good to look into the correlation between boundary strength and replication timing.

Comment3.5: We have completed this analysis using the new 6 boundary classes below.

First, below is the $-10 \log_2(\text{pvalue})$ of the **Early** IZ enrichment at our 6 boundary classes (y-axis) and the insulation score metric of boundary strength on the x-axis. As predicted from our new Figures, we see that the pvalue is greatest with the class 1 boundary farthest on the right with the strongest insulation.

Second, below again is the $-10 \log_2(\text{pvalue})$ of the **Late** IZ enrichment at our 6 boundary classes (y-axis) and the insulation score metric of boundary strength on the x-axis. As predicted from our new Figures, we see that the pvalue is now greatest with the class 6 boundary farthest on the left with the weakest insulation.

Overall – as Reviewer 3 rightly predicted, we indeed we see that the enrichment of early IZs is strongly correlated to the strength of the boundary. We also see that the enrichment of late IZ are strongly anti-correlated to the strength of the boundary.

4) Figure 2: I don't see a difference between class 3 and 6 regarding the presence of a corner-dot. In both classes this is present with a similar strength, and I don't understand the explanation in the text. If none of the individual sites show an enrichment at the corner, then how can an aggregate analysis show a corner-dot? Did the authors run an APA on these classes and see a difference in intensity?

Otherwise it is a bit odd to call it dot-less. Or does the differences between the classes lie in the boundary strength? Then surely this should be the way to classify?

Comment3.6: We have completely overhauled the boundary classes to focus specifically on only the 2 structural boundary classes already established in the literature: adjacent Corner-Dot TADs/subTADs on one or both sides (Dot boundaries) or (2) adjacent Dotless TADs/subTADs on both sides (Dotless boundaries), and then we immediately focus in Figure 1 on further stratifying Dot and Dotless boundaries by those that colocalize with (i) two or more high-density of CTCF/cohesin binding sites pointed in complex divergent/convergent orientations, (ii) one or more CTCF/cohesin binding sites pointed in tandem orientation, (iii) 0 CTCF-bound motifs (see below excerpt illustrating the 6 boundary classes used in Figure 1/2/3). We focus the rest of the paper on understanding the replication IZ patterns and how they change across a range of local and global structural perturbations. We provide APAs throughout new Figures 1-2 to ensure the classes are clear structurally and due to the molecular signatures of CTCF, cohesin, and motif orientation.

5) Figure 5: The blue rectangles indicating initiation zones run through the ChIPseq tracks, but do not at both sites touch CTCF/cohesin peaks. This does not correspond with the conclusions. Please explain.

Comment3.7: We have now provided all new figures of examples in the main Figures 1-4 as well as Supplementary Figures 1-5 to ensure the localization of CTCF+cohesin and motif orientation at each class of boundary and early/late IZs is clear. In our first submission, we provided heatmaps at inappropriate suboptimal zoom levels that were not amenable to any human clearly seeing the annotations, as the Reviewer rightly highlights. It is a tremendous challenge to show the hierarchy of large 10+ Mb-scale folding features all the way down to 100 bp CTCF binding sites. We worked hard in this submission to optimize the zoom levels to strike a balance that will allow readers to see large scale 3D genome and small-scale linear genome features.

Minor points:

- The annotation within the Hi-C maps is so large that the data in some cases has become invisible. Take for example the annotation of the corner-dots in figure 5a, which de facto mask everything. Or e.g. in figure 2a, the shape of the TAD of the "second 1st wave" is invisible in the mirror-image due to all the annotations.

Comment3.8: We have revised the Hi-C plots in all figures to address these important points in the new Figures. While the annotations are on one side of the

heatmap (upper triangle), the annotations are left off of the lower side of the heatmap (lower triangle) and the reader can use heatmap symmetry to assess the underlying features corresponding to our annotations.

- The use of the phrase 'no compartment' is confusing. A TAD is either in the A or B compartment.

Comment3.9: We have completely removed this from the paper.

Reviewer Reports on the First Revision:

Referees' comments:

Referee #1 (Remarks to the Author):

The revised manuscript is certainly much easier to read and the authors have made a good job of highlighting the key, more interesting points. They have also clarified the conclusions on replication timing and addressed, at least in the text, my concerns. However, there is one fundamental question that still puzzles me. If cohesin have such fundamental role in dictating the positioning of early, highly efficient IZs, how can cohesin depletion have such a limited and local effect on DNA replication timing overall? This is really counterintuitive. The authors should discuss this point much more in detail.

In this new version of the manuscript there are a number of places where I do not agree with the statements in the text. The paper needs to be revised to ensure that the conclusions reflect more closely the results. Potentially, revising and unifying the nomenclature of the different categories of CTCF motifs, as indicated in point 2, could clear the confusion and make the conclusions clearer and more correspondent to the data? This revision is essential to ensure that the claims are fully substantiated and the overall conclusions are clear and stand.

1. In the abstract the sentence : “By contrast, low-efficiency IZs localise to weak boundaries DEVOID of CTCF...”. The data do not show this. In Fig. 1b the only dotless boundaries significantly enriched for IZs are the 1+CTCF+cohesin. In addition, in Supp. Fig. 6 late IZs show correlation with 1+CTCF+cohesin as much as with 0CTCF+cohesin. Actually, Fig. 1b and Supp. Fig. 6 are not really saying the same.

2. On page 7: “By contrast, Dotless boundaries co-localise with late replicating IZs were most often devoid of CTCF or anchored by one CTCF motif (Fig. 1c).” It is very confusing what “one CTCF motif means”. In fig. 1C, on the bar graph is indicated that the 4 categories are : complex direction, tandem direction, SINGLE DIRECTION, no motif. There is no indication of SINGLE MOTIF. The figure legend is very unclear too. The interpretation of the role of a single motif or multiple single direction motifs could be different. It needs clarity. I think the single direction is in other figures referred to as “motif tandem”. The nomenclature should be uniform.

3. Again, in page 7: “By contrast, low efficiency IZs firing in late S-phase were depleted of Dot boundaries and significantly enriched at Dotless boundaries marked by zero CTCF+Cohesin sites (Fig. 1d-e, Supplementary Figure 7, Supplementary Methods).” This is again incorrect. In supplementary Figure 7 it there are 320 Dotless boundaries with CTCF motif tandem and 346 Dotless boundaries with no CTCF significantly enriched for late IZs.

4. Fig. 2c: the authors in the text again claim that “low-efficiency/Late IZs were enriched specifically at dotless boundaries devoid of CTCF in wildtype HCT116 cells”. However, in the figure, there are 511 dotless boundaries no CTCF, but there are 103+350 dotless boundaries with CTCF motifs either complex or tandem. I don't see the enrichment, as for hES.

5. Second paragraph on page 11: “Upon ablation of cohesin-mediated boundaries (Figures 2b, 2d, Supplementary Figure 11) we observe severe disruption of high efficiency, early S-phase IZs...”. This sentence is misleading. The dramatic effect is on class 1, less evident in class 2 and non-existing in

class 3.

6. Second paragraph on page 11: “We also noticed that low efficiency IZs shift dramatically to replicating at the end of S-phase (fractions 14-16) upon cohesin loss at dotless boundaries depleted of CTCF.” I really cannot see this dramatic shift in Fig. 2d, actually I cannot see any shift at all.

7. 5th row from the bottom, page 11, should be so modified: “Together, our ensemble and single-molecule IZ data demonstrate that disruption of cohesin-mediated loops during G1 severely alter the genomic placement where origins or clusters of origins fire during EARLY S-phase”.

8. In figure 3d I cannot see a difference in class I for WAPL KD, I see it for class 2, with a shift towards later replication.

Minor points:

9. A paper in BioRxiv shows that CTCF binding is depleted in the B compartment, (<https://doi.org/10.1101/2021.08.05.455340>). It would be interesting if the authors would discuss this parallel and how this could potentially influence the different proportion between classes 1 and 2 and classes 4 and 5 in Fig. 1c and d

10. Page 11, 4th row: “This pattern”. It is very unclear what pattern the authors refer to. Probably they mean to say: the enrichment of high efficiency/early IZs only happens at boundaries that co-localize with cohesin (Supplementary Figure 12).

11. Supplementary Figure 10: I don't seem to find a FACS to measure BrdU incorporation in the wildtype. How can I judge if Rad21 depletion has any effect? Sorry if I missed it and it is somewhere else.

12. Fig. 3c: in the middle of page 13 it says “It is particularly noteworthy that the most significant gain-of-looping..”. In what sense “most significant”. Statistically the gain-of-looping in class 2 is more significant.

Referee #2 (Remarks to the Author):

I appreciate the effort made by the authors on addressing my specific concerns. Specially the demanding analysis on different loop callers having or not an effect in their final findings. As expected, calling loops results in limited reproducibility between different methods/parameters. However, it was ensuring that the conclusions of the downstream analysis are maintained.

In summary, I have no additional concerns or requests for the current version of the manuscript.

Referee #3 (Remarks to the Author):

The manuscript ‘Cohesin-mediated loop anchors confine the location of human replication origins’ has been rewritten following the recommendations of the referees. In my opinion these edits have strongly improved the manuscript. The simplification of the different classes makes the manuscript much easier to read, and the finding that complex boundaries are involved in the positioning of replication initiation locations provides important new insights for the field. The extension of the

experimental section, including WAPL depletion experiments, also strengthens the whole.

Only a few points remain:

The annotation in figure 1 remains sub-optimal. Look for example at the green circles that are supposed to highlight corner dots. In the actual figure, these are just unclear green blobs.

In figure 3 the authors now include their experiments in a WAPL-depleted setting. In correspondence with earlier work, the authors find longer loops, which interestingly are mainly found in the complex boundary class.

The differences however are quite hard to spot by eye, and even were invisible when I printed out the figures. I expect that the addition of differential plots could be helpful to make figure 3 clearer.

Apart from these minor comments, I would support publication.

Author Rebuttals to First Revision:

Overview for Reviewers

Thank you to all three Reviewers for their constructive feedback on our revised manuscript. Please find below our general outline of changes in this resubmission:

- We have clarified language to reflect that low-efficiency / late initiation zones localize to weak Dotless boundaries.
- We have clarified the nomenclature of the 6 boundary classes to reflect their motif composition more precisely.
- We revised multiple figure panels to increase clarity
- We provide additional data analysis below in responses to questions about replication timing

We appreciate the time and effort the reviewers have invested to help us improve our work. Our responses to each comment are included in line with their comments below in blue font with text from the new manuscript in quotations.

Best,

Jennifer E. Phillips-Cremins,
on behalf of all authors in the Chen, Cremins, Dekker, Gilbert, and Yue labs

Referees' comments:

Referee #1 (Remarks to the Author):

The revised manuscript is certainly much easier to read and the authors have made a good job of highlighting the key, more interesting points. They have also clarified the conclusions on replication timing and addressed, at least in the text, my concerns. However, there is one fundamental question that still puzzles me. If cohesin have such fundamental role in dictating the positioning of early, highly efficient IZs, how can cohesin depletion have such a limited and local effect on DNA replication timing overall? This is really counterintuitive. The authors should discuss this point much more in detail.

Comment 1.1: We would first like to thank again the Reviewer for their time and detailed attention to our revised manuscript.

To address the Reviewer's important question, we generated a histogram of the fraction of WT vs. RAD21 KD genomic bins replicating at different times during S phase, using two-fraction Repli-seq and a $\log_2(E/L)$ ratio (**Rebuttal Figure 1**). We find a global decrease in dynamic range of replication timing in Rad21 knock-down HCT116 compared with wild type (**Rebuttal Figure 1a**), consistent with origin de-localization reducing the extremes of earliest and latest replication timing by increasing the heterogeneity of distances of bins from initiation sites.

Rebuttal Figure 1. (A) $\log_2(E/L)$ two-fraction Repli-seq dynamic range in wild type vs. Rad21 knock-down HCT116 cells. (B) Simulation of 16-fraction Repli-seq data with 100kb, 300kb, 1 Mb, and 3Mb widening of IZs.

In addition, we simulated 16-fraction Repli-seq for both wild type (WT) and Rad21 knock-down (KD) HCT116 via progressively more severe widening of WT initiation zones (IZs) (**Rebuttal Figure 1b**). We simulated the WT initiation features by assuming genomic loci at WT IZs have a probability of initiating equal to 1. In each ensuing S phase fractions, the neighboring 50 kb genomic bins are replicated passively as a result of replication forks moving left- and right-ward. When two forks merge, they form a termination zone represented as a valley. To simulate the

Rad21 KD replication features, we used our experimental Rad21 KD IZ width (**Supplementary Figure 11c**) to determine Rad21 KD IZs are on average 30 kb wider than WT IZs. In our simulation we widened the WT IZs by 100 kb, 300 kb, 1 Mb, and 3 Mb to generate simulated 16-fraction Repli-seq. In our simulation we observed the overall replication timing features are generally stable up to IZ widths 33x larger (1 Mb) than those observed in our Rad21 KD experiment. To disrupt replication timing, IZ width greater than 3 Mb wider than the WT is required (**Rebuttal Figure 1b**).

Results from the simulation and $\log_2(E/L)$ RT profile are consistent with our model in which delocalized/diffused replication initiation factors spread in the absence of Rad21 and subsequently initiate replication over wider DNA segments causing a diffused and plateaued appearance of the IZs. The diffused and plateaued appearance of IZs as simulated in **Rebuttal Figure 1b** would not cause a dramatic localized shift such as an early to late shift in $\log_2(E/L)$ RT profile as these peaks are still replicated early in S phase. Rather this diffusion leads to widened peaks, thus culminating in a loss of dynamic range as seen in **Rebuttal Figure 1a**.

In this new version of the manuscript there are a number of places where I do not agree with the statements in the text. The paper needs to be revised to ensure that the conclusions reflect more closely the results. Potentially, revising and unifying the nomenclature of the different categories of CTCF motifs, as indicated in point 2, could clear the confusion and make the conclusions clearer and more correspondent to the data? This revision is essential to ensure that the claims are fully substantiated and the overall conclusions are clear and stand.

Comment 1.2: We appreciate the Reviewer's points and we have made changes in the text as detailed below. In the resubmission we also rename the six boundary classes and present revised labels in Figures 1 and 2 for increased clarity.

In the abstract the sentence : "By contrast, low-efficiency IZs localise to weak boundaries DEVOID of CTCF...". The data do not show this. In Fig. 1b the only dotless boundaries significantly enriched for IZs are the 1+CTCF+cohesin. In addition, in Supp. Fig. 6 late IZs show correlation with 1+CTCF+cohesin as much as with 0CTCF+cohesin. Actually, Fig. 1b and Supp. Fig. 6 are not really saying the same.

2. On page 7: "By contrast, Dotless boundaries co-localise with late replicating IZs were most often devoid of CTCF or anchored by one CTCF motif (Fig. 1c)." It is very confusing what "one CTCF motif means". In fig. 1C, on the bar graph is indicated that the 4 categories are : complex direction, tandem direction, SINGLE DIRECTION, no motif. There is no indication of SINGLE MOTIF. The figure legend is very unclear too. The interpretation of the role of a single motif or multiple single direction motifs could be different. It needs clarity. I think the single direction is in other figures referred to as "motif tandem". The nomenclature should be uniform.

Comment 1.3: We have now revised **Figure 1b, 1c, 1d** and the captions for **Figure 1** and **Supplementary Figure 6**, as well as text for **Figure 1**, to fix the format and ensure the figure conclusions are clear. We modified the abstract to read: "By contrast, low-efficiency IZs localize to weak dotless boundaries." And have removed the 'devoid of CTCF' claim. We have revised **Figure 1b** to ensure the conclusion is much more clearly labeled compared to the previous version. **Figure 1b** explores the number of co-bound CTCF+cohesin sites or the number of

cohesin only sites per boundary. The new format of **Figure 1b** - as well as the new caption and text - are revised to ensure it is clear that the questions we asked in **Figures 1c, d, e**, and **Supplementary Figure 6** are completely different. On page 6 we add the following text to explain the result:

“””

We observe a significantly higher density of co-bound CTCF+cohesin binding sites at Dot boundaries overlapping early replicating IZs compared to those that do not overlap any IZs, respectively (**Figure 1b, Supplementary Tables 5-6**). We also examined cohesin-only binding sites, as they can earmark CTCF-independent enhancer-promoter interactions^{9, 10, 23}, but we did not see a notable difference in number across Dot vs. Dotless TAD/subTAD boundaries (**Figure 1b**). Together, our data indicate that boundaries co-localizing with human early S phase IZs exhibit enriched occupancy of CTCF+cohesin co-bound sites, but not cohesin alone, thus confirming and significantly expanding on observations in previous reports linking cohesin generally to a small subset of replication origins in *Drosophila*²⁶ and humans²⁷.

“”””

We also reformatted **Figure 1c** to drive clarity. This figure is a different claim than **Figure 1b** – as the goal is to explore the *orientation of CTCF motifs* in Dot and Dotless boundaries – whereas **Figure 1b** is assessing the *number CTCF+cohesin binding sites*. We make it very clear in the legend that we are looking at boundaries with four types of motif scenarios co-bound by CTCF+cohesin – those that are in complex divergent/convergent orientations, those that are 2+ and tandem orientation, those that are single motifs only, and finally those without any motifs.

On page 6 we revise the following text to ensure the focus on motif orientation is clear:

“””

Recent reports have uncovered that convergently-oriented CTCF motifs anchor long-range looping interactions formed by cohesin-mediated extrusion^{11, 23, 28-30}. The signature of CTCF+cohesin bound motifs in complex orientation was further enriched at dot boundaries colocalized with early replicating IZs versus those without IZs. By contrast, nearly all Dotless boundaries have only one or no CTCF+cohesin bound motifs (**Figure 1c**). This pattern was further enriched at dot boundaries co-localized with early replicating IZs.

“””

Finally, we clarify that the analyses shown in **Figures 1d-e** and **Supplementary Figure 6c** are completely different than **Figures 1b+c**. We clarify here that our conclusions in **Figure 1d** and **Supplementary Figure 6** are built on *the aggregate Repli-seq signal* at boundaries after stratifying 6 types of boundaries – and not based on the numbers of boundaries in each class. As now stated in the text, we stratified Dot (Class 1-3) and Dotless (Class 4-6) boundaries into those (i) localized with CTCF+cohesin bound motifs in complex orientation (Classes 1, 4), (ii) tandem or single-motif orientation (Classes 2, 5), or (iii) no bound motifs (Classes 3, 6) (**Figure 1d**). The main message from **Figure 1d** is conveyed in the aggregate Repliseq plots under the heatmaps. There is a strong clear enrichment of Repliseq signal indicative of early S phase IZs in the center of the plots at the boundaries predominantly at Class 1 boundaries. We also revisited our observations of the late IZs given the Reviewer’s valid points. In studying the Repli-seq

signal it is clear that late IZs are enriched (see visual Repliseq pileups) directly at the Class 5 and 6 Dotless boundaries with either tandem + single CTCF motifs (Class 5) or no bound CTCF motifs (6). We revised the text in the caption, main, and abstract to reflect this point, as exemplified by the new claims on p. 7:

“”””””

Consistent with our qualitative observations, high-efficiency IZs firing in early S phase were significantly enriched at Dot boundaries marked by CTCF+cohesin binding sites in complex orientations compared to a null distribution of randomly placed IZs matched to the same A/B compartment distribution (Class 1, **Figures 1d-e, Supplementary Figure 6b-d, Supplementary Methods**). By contrast, low-efficiency IZs firing in late S phase were depleted at Dot boundaries and significantly enriched at Dotless boundaries with tandem+single CTCF+cohesin bound motifs or no bound motifs (Classes 5 and 6, **Figures 1d-e, Supplementary Figure 6b-d, Supplementary Methods**).

“””””

Finally, in the caption and in the revised **Figure 1d** schematic, we make textual changes to clarify that we combine “one motif” and “tandem orientation” motifs into a single group “tandem + single orientation” moving forward in the manuscript. The reason for this is that both 2+ tandem and single motifs are pointing in the same direction, and they biologically looked very similar in our analysis of the Repli-seq signal. To keep the manuscript high level and clear for a broader readership, we believe it is best to merge these two groups – and it is labeled now.

3. Again, in page 7: “By contrast, low efficiency IZs firing in late S-phase were depleted of Dot boundaries and significantly enriched at Dotless boundaries marked by zero CTCF+Cohesin sites (Fig. 1d-e, Supplementary Figure 7, Supplementary Methods).” This is again incorrect. In supplementary Figure 7 it there are 320 Dotless boundaries with CTCF motif tandem and 346 Dotless boundaries with no CTCF significantly enriched for late IZs.

Comment 1.4: We clarify above in Comment 1.3 and here that our conclusions in **Figure 1d, Figure 2, and Supplementary Figure 7** are built on *the aggregate Repli-seq signal* at boundaries after stratifying six types of boundaries – and not based on the numbers of boundaries in each class.

We have revised the **Figure 1d** and **Figure 2** schematics to ensure that the six boundary classes are clear – and ensure the readers focus on the Repli-seq signal and not the numbers of each boundary. There is a strong clear enrichment of Repliseq signal indicative of early S phase IZs in the center of the plots at the boundaries predominantly at Class 1 boundaries in both **Figure 1d** and **Figure 2**.

We also revisited our observations of the late IZs given the Reviewer’s valid points. In studying the Repli-seq signal it is clear that late IZs are enriched (see visual Repliseq pileups) directly at the Class 5 and 6 Dotless boundaries with either tandem + single CTCF motifs (Class 5) or no bound CTCF motifs (6). We revised the text in the caption, main, and abstract to reflect this point, as exemplified by the new claims on p. 7:

“””””

Consistent with our qualitative observations, high-efficiency IZs firing in early S phase were significantly enriched at Dot boundaries marked by CTCF+cohesin binding sites in complex orientations compared to a null distribution of randomly placed IZs matched to the same A/B compartment distribution (Class 1, **Figures 1d-e, Supplementary Figure 6b-d, Supplementary Methods**). By contrast, low-efficiency IZs firing in late S phase were depleted at Dot boundaries and significantly enriched at Dotless boundaries with tandem+single CTCF+cohesin bound motifs or no bound motifs (Classes 5 and 6, **Figures 1d-e, Supplementary Figure 6b-d, Supplementary Methods**).

“””””

4. Fig. 2c: the authors in the text again claim that “low-efficiency/Late IZs were enriched specifically at dotless boundaries devoid of CTCF in wildtype HCT116 cells”. However, in the figure, there are 511 dotless boundaries no CTCF, but there are 103+350 dotless boundaries with CTCF motifs either complex or tandem. I don’t see the enrichment, as for hES.

Comment 1.5: Thank you very much to Reviewer 1 for helping us see the areas where our former manuscript version was not clear. We hope all of the figure and text revisions hit the mark. We provide the clarification to this issue in both Comments 1.3 and 1.4. There are three important points. First, we are not making claims based on the numbers in each boundary class, as the Reviewer states above. The numbers simply represent the numbers of boundaries with each molecular signature and cannot be used to interpret enrichment. We apologize if our Figure design caused the issue – and we have revised the schematics in **Figures 1d, Figures 2, and Supplemental Figures** to ensure each boundary class is clear. Second, the enrichment of early IZs is observed in the aggregate Repliseq signal itself. Please note that early IZs show a clear and precise and specific pattern of enrichment at only Class 1 boundaries in both hES and HCT116 cell lines. This qualitative pattern is clearly confirmed quantitatively with **Figure 1e** with our statistical test. Third, the enrichment of late IZs are also observed in the aggregate Repliseq signal itself, not the number of boundaries. In **Figure 2c** it is clear that late IZs exhibit focal signal in the Repliseq at Dotless boundaries (Classes 4-6). We agree this pattern is not only at Dotless boundaries devoid of CTCF and we have edited the text to communicate this in HCT116 as in p. 10 (below):

“””””

As in hES cells, we observed that 16-fraction Repli-seq data exhibits focal enrichment of high-efficiency/Early IZs specifically at Dot boundaries marked by CTCF+cohesin co-bound motifs in complex orientation in wild type HCT116 cells (Class 1, **Figure 2c**). Enrichment of high-efficiency/Early IZs only occurs at boundaries that colocalize with cohesin (**Supplementary Figure 10**). Moreover, as in hES cells, low-efficiency, Late IZs were enriched at weak Dotless boundaries in wild type HCT116 cells (**Figure 2c**).

“”””

Finally, we do want to emphasize that nearly the entire manuscript focuses on functional experiments exploring the enrichment and link between Class 1 boundaries and early S phase IZs. We clean up the wording and figures as the Reviewer wisely suggests, but the Late IZs

addressed in Comments 1.1-1.5 are only a very small part of the story. Thank you again for your insight in this matter.

5. Second paragraph on page 11: “Upon ablation of cohesin-mediated boundaries (Figures 2b, 2d, Supplementary Figure 11) we observe severe disruption of high efficiency, early S-phase IZs...”. This sentence is misleading. The dramatic effect is on class 1, less evident in class 2 and non-existing in class 3.

Comment 1.6: We have made a textual change on p. 11 to clarify that high efficiency, early S-phase IZs are disrupted specifically at Class 1 boundaries.

“””

Upon ablation of cohesin-mediated boundaries (**Figures 2b, 2d, Supplementary Figure 9**), we observe severe disruption of high-efficiency early S phase IZs specifically *at Class 1 boundaries*, as evidenced by diffuse and de-localized Repli-seq signal (*Class 1, Figure 2c, 2e*).

“”””

6. Second paragraph on page 11: “We also noticed that low efficiency IZs shift dramatically to replicating at the end of S-phase (fractions 14-16) upon cohesin loss at dotless boundaries depleted of CTCF.” I really cannot see this dramatic shift in Fig. 2d, actually I cannot see any shift at all.

Comment 1.7: Thank you for raising this issue. If the Reviewer compares **Figure 2c** Repliseq in wild type HCT116 to **Figure 2e** Repliseq in cohesin knock-down HCT116 --- the Repliseq is aggregated at fractions 9-14 in Classes 4/5 Dotless boundaries (**Figure 2c**) and shift down to fractions 14-16 in Classes 4/5 Dotless boundaries (**Figure 2e**). Moreover, for Class 6 Dotless --- the Repliseq is aggregated at fractions 12-16 (**Figure 2c**) and shift down to fractions 14-16 (**Figure 2e**). We also see the late S phase Repliseq signal spread laterally upon cohesin knock-down at Classes 4-6 boundaries. We have modified text on p. 11 to clarify that we see a shift in the Repli-seq signal itself in Class 4, 5, and 6 Dotless boundaries.

“””

We also noticed that low-efficiency IZs shift to replicating at the end of S phase (fractions 14-16) at Dotless boundaries upon cohesin knock-down (Classes 4-6, **Figures 2c+2e, Supplementary Figure 11**).

“””

7. 5th row from the bottom, page 11, should be so modified: “Together, our ensemble and single-molecule IZ data demonstrate that disruption of cohesin-mediated loops during g1 severely alter the genomic placement where origins or clusters of origins fire during EARLY s-phase”.

Comment 1.8: We have modified the text as suggested on p. 11:

“””

Together, our ensemble and single-molecule IZ data demonstrate that disruption of cohesin-

mediated loops during G1 severely alters the genomic placement where origins or clusters of origins fire during early S phase.

8. In figure 3d I cannot see a difference in class I for WAPL KD, I see it for class 2, with a shift towards later replication.

Rebuttal Figure 2. We have pasted Figure 4d here with the revised boundary schematic inspired by Reviewer Comments 1.1-1.7.

Comment 1.9: We have modified the boundary labeling scheme to ensure the classes are clear in **Figure 4d** – and we have pasted **Figure 4d** above for the Reviewer’s ease as **Rebuttal Figure 2**. Our conclusions are based on the early fraction Repliseq enrichment (focal peak in the center between -750 kb and + 750 kb) at Class 1 boundaries (second row, wild type Repliseq, Class 1) which becomes more precise/narrow upon WAPL knock-down (third row, Repliseq, Class 1) and more diffuse upon cohesin knock-down (first row, Repliseq, Class 1, as shown in **Figure 2**). In the previous version, we believe our suboptimal boundary class labeling scheme has led the readers eyes away from the Repliseq pattern. We take responsibility for this, and we hope these improvements allow the patterns in WAPL KD at Class 1 boundaries to now take center stage. There is only a slight narrowing around mid S phase at Class 2 boundaries upon WAPL KD, and it is not as profound at Class 1.

To ensure this pattern is not just visual, in **Figure 4e** we demonstrate that the IZs called in wild type HCT116 became narrower in WAPL knock-down at both Class 1 and 2 boundaries, consistent with the qualitative result described above and in **Figure 4d**.

We have ensured the conclusion reflects the data on p. 13:

“””””

At Class 1 boundaries, we observe that Early IZs become significantly narrower upon WAPL

knock-down (**Figures 3d-e, Supplementary Table 17**). It is striking that IZs tighten and refine upon gain-of-looping in the WAPL knock-down condition at the same boundaries where IZs grow more diffuse upon cohesin knock-down (**Figure 3a-b, Supplementary Figure 11**).

Minor points:

9. A paper in BioRxiv shows that CTCF binding is depleted in the B compartment, (<https://doi.org/10.1101/2021.08.05.455340>). It would be interesting if the authors would discuss this parallel and how this could potentially influence the different proportion between classes 1 and 2 and classes 4 and 5 in Fig. 1c and d

Comment 1.10: We have revised the text on p. 6-7 to state the following, based on the improved **Figure 1c**. **Figure 1c** is also pasted below as **Rebuttal Figure 3** for Reviewer ease.

“””

By contrast, nearly all Dotless boundaries have only one or no CTCF+cohesin bound motifs (**Figure 1c**), which is consistent with the recent report suggesting that CTCF density is lower in the B compartment where Dotless boundaries most often reside (<https://doi.org/10.1101/2021.08.05.455340>).

“””

Rebuttal Figure 3. We have pasted the new revised/relabelled Figure 1c here for clarity.

We also note that we explicitly account for the A/B compartment distribution in our statistical test for IZ enrichment at boundaries, as in text on p. 7:

“”””

We note that our null distribution was matched to real IZs by their A/B compartment distribution, such that the enrichment reflects a strong localization at boundaries transcending the known link between early and late replicating domains and A and B compartments, respectively (**Supplementary Methods**).

“”””

10. Page 11, 4th row: “This pattern”. It is very unclear what pattern the authors refer to. Probably they mean to say: the enrichment of high efficiency/early IZs only happens at boundaries that co-localize with cohesin (Supplementary Figure 12).

Comment 1.11: We have adjusted the text to clarify (now bottom of p.10). Supplementary figure numbers have been adjusted because we combined figures according to Nature’s formatting requirements.

“”

Enrichment of high-efficiency/Early IZs only occurs at boundaries that colocalize with cohesin (**Supplementary Figure 10**).

“”

11. Supplementary Figure 10: I don’t seem to find a FACS to measure BrdU incorporation in the wildtype. How can I judge if Rad21 depletion has any effect? Sorry if I missed it and it is somewhere else.

Comment 1.12: We apologize for any confusion. We have analyzed BrdU IP efficiency in **Supplementary Figure 8f** where we see consistent IP efficiency across conditions and fractions. Additionally, the specific FACS measurements mentioned above also based on BrdU have been previously published by Oldach & Nieduszynski (<https://www.ncbi.nlm.nih.gov/pmc/articles/PMC6471042/>) using the same cell line background and synchronization conditions used in the present study.

We provide the data (**from Oldach & Nieduszynski**) below as **Rebuttal Figure 4** – demonstrating a flow cytometry analysis of BrdU uptake and DNA content in the same cell line and synchronization scheme that we employ to knock-down Rad21&WAPL. Consistent with our results in **Supplementary Figure 8f**, the Nieduszynski study also shows minimal clear effect of Rad21 knock-down on BrdU incorporation.

To ensure this clarity for other readers we have added an additional citation to Oldach & Nieduszynski (<https://www.ncbi.nlm.nih.gov/pmc/articles/PMC6471042/>) when these results are mentioned in the text and again in the methods.

Rebuttal Figure 4: from Oldach & Nieduszynski, Supplementary Figure 3a [Figure redacted]

12. Fig. 3c: in the middle of page 13 it says “It is particularly noteworthy that the most significant gain-of-looping..”. In what sense “most significant”. Statistically the gain-of-looping in class 2 is more significant.

Comment 1.13: In the previous version, we mistakenly annotated the figure in a way that suggested Class 2 was more significant. This was a typo. We have corrected the statistical annotation in **Figure 3c**, and have revised the text at the bottom of p. 11 as follows:

“””

We observed that the gain-of-looping phenotype upon WAPL knock-down occurs most strongly at Dot boundaries with complex CTCF motif orientation (Class 1, **Figure 3c**).

“””

Referee #2 (Remarks to the Author):

I appreciate the effort made by the authors on addressing my specific concerns. Specially the demanding analysis on different loop callers having or not an effect in their final findings. As expected, calling loops results in limited reproducibility between different methods/parameters. However, it was ensuring that the conclusions of the downstream analysis are maintained. In summary, I have no additional concerns or requests for the current version of the manuscript.

Comment 2.1: We want to thank the Reviewer for their time and comments as they have helped us make this a stronger manuscript.

Referee #3 (Remarks to the Author):

The manuscript ‘Cohesin-mediated loop anchors confine the location of human replication origins’ has been rewritten following the recommendations of the referees. In my opinion these edits have strongly improved the manuscript. The simplification of the different classes makes the manuscript much easier to read, and the finding that complex boundaries are involved in the positioning of replication initiation locations provides important new insights for the field. The extension of the experimental section, including WAPL depletion experiments, also strengthens the whole. Only a few points remain:

The annotation in figure 1 remains sub-optimal. Look for example at the green circles that are supposed to highlight corner dots. In the actual figure, these are just unclear green blobs.

Comment 3.1: Thank you. We have updated the loop annotation in **Figure 1a** to accurately reflect that loops are annotated using green circles. We have lowered the line width of the green circles, and we have also modified heatmap color scales (**Rebuttal Figure 5**). This change has been reflected throughout the manuscript and supplement.

Rebuttal Figure 5. Improvement in heatmap loop annotation, legend, and color scale in Figure 1a and other supplementary figures.

In figure 3 the authors now include their experiments in a WAPL-depleted setting. In correspondence with earlier work, the authors find longer loops, which interestingly are mainly found in the complex boundary class.

The differences however are quite hard to spot by eye, and even were invisible when I printed out the figures. I expect that the addition of differential plots could be helpful to make figure 3 clearer.

Comment 3.2: We have optimized the heatmap color scale and added more detailed blue arrows to **Figure 3a-b** to visually clarify where we see gains of long loops. We also tried differential heatmaps but after many attempts, we found that the current color scale and arrow annotation highlighted the gained loops most clearly in our hands.

Apart from these minor comments, I would support publication.

Comment 3.3: Thank you again for your time and suggestions to improve the work.

Reviewer Reports on the Second Revision:

Referees' comments:

Referee #1 (Remarks to the Author):

The authors have addressed and clarified all the points raised. The new figures and the clean up of the classification make it much easier to read. I found the Rebuttal Figure 1 very interesting and instructive. I would suggest do add it to the supplemental information. The approach adopted to address the question is very interesting. I am looking forward to see the paper published.

Author Rebuttals to Second Revision:

Referees' comments:

Referee #1 (Remarks to the Author):

Referees' comments:

Referee #1 (Remarks to the Author):

The authors have addressed and clarified all the points raised. The new figures and the clean up of the classification make it much easier to read. I found the Rebuttal Figure 1 very interesting and instructive. I would suggest do add it to the supplemental information. The approach adopted to address the question is very interesting. I am looking forward to see the paper published.

We thank the reviewer for their positive comments on our revision, and for their time investment in providing feedback which genuinely helped us improve the manuscript at each step. We have elected to keep the final simulation in the rebuttal. Simulation of 16 fraction Repli-seq data is a new open-ended and sparsely explored area and would warrant a full additional figure of exploration and discussion. Given the space constraints, we could not fully develop and add this new idea at the quality and depth required for publication.